# The global speciation continuum of the cyanobacterium *Microcoleus*

Aleksandar Stanojković [1], Svatopluk Skoupý[1], Hanna Johannesson[2,3] & Petr Dvořák [1] ✉

Speciation is a continuous process driven by genetic, geographic, and ecological barriers to gene flow. It is widely investigated in multicellular eukaryotes, yet we are only beginning to comprehend the relative importance of mechanisms driving the emergence of barriers to gene flow in microbial populations. Here, we explored the diversification of the nearly ubiquitous soil cyanobacterium *Microcoleus*. Our dataset consisted of 291 genomes, of which 202 strains and eight herbarium specimens were sequenced for this study. We found that *Microcoleus* represents a global speciation continuum of at least 12 lineages, which radiated during Eocene/Oligocene aridification and exhibit varying degrees of divergence and gene flow. The lineage divergence has been driven by selection, geographical distance, and the environment. Evidence of genetic divergence and selection was widespread across the genome, but we identified regions of exceptional differentiation containing candidate genes associated with stress response and biosynthesis of secondary metabolites.

A complex interplay of evolutionary forces generates and governs differentiation within and among groups of organisms, leading to species emergence[1,2]. Recent advances in population and comparative genomics have greatly facilitated the articulation of the evolutionary forces shaping speciation and key genes involved in environmental adaptation[3]. However, understanding the evolutionary processes driving differentiation and the emergence of new species in prokaryotes is still hindered by ambiguous species concepts for microbes[4], their extensive genetic and phenotypic diversity[5], and the highly heterogeneous ecological niches they can occupy[6].

Speciation is a continuous process representing any stage of ecological and genetic differentiation of populations[7] that are ultimately identified as species. Stankowski & Ravinet[8] posited the speciation continuum as a 'continuum of reproductive isolation' rooted in the biological species concept. While this notion is well-suited for sexually reproducing eukaryotes, there is no equivalent definition for the speciation continuum in prokaryotes. Even though microbes do not possess the necessary machinery for canonical sexual reproduction, gene flow takes place at varying frequencies between groups on different levels of genomic divergence. This leaves detectable traces in genetic material akin to sexual reproduction in eukaryotes[9–11].

Accordingly, many authors regard extensively recombining bacteria as quasisexual[12]. Building upon Stankowski & Ravinet's definition of the speciation continuum, it can be adapted to microbes and defined as a continuum of barriers to gene flow, where gene flow encompasses the transfer of DNA material realized by homologous recombination (HR). Several studies have already shown that the evolutionary forces contributing to divergence, such as genetic and ecological isolation and selection, could aid in delineating groups of closely related microbes. This fits into the concept that species are groups of individuals and populations genetically and ecologically isolated from other such groups (e.g., *Vibrio*[13]; *Sulfolobus*[14]; *Microcystis*[15]; *Laspinema*[16]).

Factors that introduce genetic changes to microbial populations include horizontal gene transfer (HGT), mutations, adaptation, and gene flow mediated by homology-(in)dependent mechanisms. The interaction between selection, HGT, HR, and genetic drift dictates the magnitude of differentiation among populations[13,17,18]. Depending on the coaction of these forces, species can be at various stages of speciation across space and time, from undergoing continuous gene flow to its complete cessation[19–21]. The former viewpoint on speciation centering on the extent of gene flow between species, which varies along a continuum, has regained attention[22]. This perspective challenges the

¹Palacký University Olomouc, Faculty of Sciences, Department of Botany, Olomouc, Czech Republic. ²Department of Ecology, Environment and Plant Sciences, Stockholm University, Stockholm, Sweden. ³The Royal Swedish Academy of Sciences, Stockholm, Sweden. ✉e-mail: p.dvorak@upol.cz

traditional view of speciation as a discrete and irreversible event[23,24]. Numerous genomic studies have tested this idea by focusing on genetic differentiation between closely related incipient species to characterize them at various stages of genetic and ecological divergence in both eukaryotes (*Anopheles* mosquitoes[25]; *Heliconius* butterflies[26]; *Helianthus* sunflowers[27]) and prokaryotes (*Vibrio*[13]; *Laspinema*[16]).

In the early stages of speciation, microbial populations may exhibit subtle genomic hallmarks of elevated differentiation localized in small genomic regions under strong divergent selection and impervious to gene flow, often referred to as 'islands of speciation'[19,28,29]. The emergence of these regions can be propelled by adaptation to various environmental factors such as light and stress (e.g., *Prochlorococcus*[30,31]; *Laspinema*[16]) or nutrient uptake (e.g., *Vibrio*[13]). As speciation progresses, genomic landscapes of divergence may become more distinct and fixed, establishing genome-wide barriers and, ultimately, genetically and ecologically distinct species[32]. Widespread genetic differentiation has been reported in the prokaryotes *Sulfolobus*[14] and *Laspinema*[16] as well as in eukaryotes like mosquitoes[33] and fruit flies[34]. Whether the speciation initiates from 'islands' or arises across the whole genome remains unresolved in eukaryotes, with authors reporting contrasting patterns among different taxa[35].

Cyanobacteria have been dominating ecosystems as primary producers for billions of years[36]. Today, the most prominent and widely distributed is *Microcoleus*, a mat-forming, filamentous cyanobacterium inhabiting diverse aerophytic and benthic habitats[37,38]. One particularly successful and abundant species is *Microcoleus vaginatus*, occurring in biological soil crusts of arid and semiarid ecosystems, which cover up to 40% of the land surface on Earth[39]. The Cenozoic rise to dominance of *M. vaginatus* 39.5 million years ago (Mya)[40] undoubtedly affected dry ecosystems and was central to the evolution of biocrust communities[41]. Thriving under unique and challenging ecosystem conditions (e.g., water scarcity, high climatic variability, low carbon storage capability), *M. vaginatus* plays a key role in terms of soil stability, primary productivity, and carbon influx[42]. Moreover, *Microcoleus*-dominated crusts improve soil fertility and moisture retention, molding microhabitats for other microbial and plant communities[43]. More than 3500 studies have focused on *M. vaginatus*, but it is actually just one of the many species along the entire continuum of species at varying stages of speciation (we queried the Google Scholar database; accessed 25th June 2023; the term used: *Microcoleus vaginatus*). In our previous study, using a global dataset of almost 500 strains based on 16S rRNA and 16S-23S ITS markers, we noticed that what has been referred to as *M. vaginatus* embodies vast genetic diversity and may consist of at least 12 lineages, which diversified due to the influence of geographical and ecological separation[44].

Here, we investigate the continuum of *Microcoleus* populations and possible genetic determinants connected with the capacity to dominate worldwide dryland ecosystems. We sequenced 202 whole genomes of multiple closely related strains and eight whole genomes of herbarium specimens deposited from 1851 to 1938. We added these to the 81 publicly available genomes and examined the genetic population structure and whether geography and environment affect differentiation within *Microcoleus*. We next explored genome-wide inter- and intraspecific diversity to provide a better understanding of the potential cause of the divergence within *Microcoleus*. Finally, we investigated the signatures of local selection and gene flow over the whole genome during adaptive divergence.

## Results

We obtained complete genome sequences for 202 *Microcoleus* strains and eight herbaria-preserved *M. vaginatus* strains from across diverse geographic and climatic settings. De novo assembly yielded 202 genomes of a final mean depth 64.89×, ranging in length from 6.39 to 9.64 Mb with mean completeness of 99.5% (minimum 91%) and 0.27

mean contamination level. Genomes of herbarium specimens had a final mean depth of 28.9× and length between 6.90 and 8.28 Mb with mean completeness of 94.5% (minimum 85.8%) and 0.43 mean contamination level (Supplementary Data 1). These genomes were used together with 81 *Microcoleus* genomes obtained from the GenBank to generate a comprehensive, global dataset of this group. We constructed three datasets for different analyses. Dataset I was used to investigate the phylogenetic positions of *Microcoleus* in the broader cyanobacterial phylogeny and consisted of 165 genomes, 36 selected from strains sequenced in this study and 129 representatives from the GenBank (accession numbers in Supplementary Data 2). Dataset II included 210 genomes sequenced in this study and 81 *Microcoleus* genomes obtained from GenBank database (accession numbers in Supplementary Data 2); wholly, 291 genomes originating from all continents except for South America (Fig. 1a). This dataset was constructed to fully encompass the wide range of ecological factors and geographic locations where this cyanobacterial group thrives, improving the ancestral area reconstruction and phylogenomic analyses while also enhancing the statistical significance of Mantel's correlation[45] and phylogenetic signal tests[46,47]. The final dataset III included 202 genomes sequenced in this study and was used for the inference of the species trees and dating analysis (with the outgroup, strain M2_D5). The outgroup was chosen as the most closely related to the *Microcoleus* clade (Supplementary Fig. 1), and it was omitted for variant calling, delimitation, pangenome, recombination, horizontal gene transfer, and population genetic analyses.

### Geography contributes to the diversification of the *Microcoleus* global continuum

The evolutionary history of selected *Microcoleus* isolates and other cyanobacteria from the public database was reconstructed using the multiple sequence alignment (MSA) with 129,335 amino acid sites (dataset I). The inferred maximum-likelihood (ML) tree revealed that our strains clustered in a monophyletic clade, with the closest sister taxa being *Microcoleus* sp. M2_D5 and *Oscillatoria* sp. PCC 6506 (Supplementary Fig. 1).

The evolutionary history of the 291 *Microcoleus* strains was reconstructed using MSA with 306,182 amino acid sites (dataset II). The ancestral geographical area reconstruction suggested a European origin for most species (Supplementary Fig. 2). The basal clade includes African and North American strains, and it had a deep split from the common ancestor of all our *Microcoleus* isolates (Fig. 1a). The phylogeographic patterns of our *Microcoleus* isolates suggest that the species found in geographical areas other than Europe are descendants of more than one expansion out of Europe. Moreover, *Microcoleus* displayed a distance-decay relationship ($r = 0.44$, $p < 0.0001$), i.e., strains tend to exhibit greater genetic similarity as the physical distance decreases. This indicates that geography represents a contributory factor to the diversification of *Microcoleus* lineages (Fig. 1b).

### *Microcoleus* comprises a global continuum of at least 13 lineages

We compared species trees inferred with three different approaches to explore the evolutionary relationships among our *Microcoleus* isolates (dataset III): (1) an ML tree from 2020 single-copy orthologues, (2) an ML tree from all single nucleotide polymorphisms (SNPs) across the genomes, and (3), an ASTRAL tree from a set of unrooted trees under the multispecies coalescent model (Supplementary Figs. 3-5). There was a discordance in the phylogenetic positions of a few strains in the ASTRAL and the SNP tree compared to the ML tree from single-copy orthologues, which could be due to either incomplete lineage sorting or gene flow. The topology of the species tree did not alter when recombination spots were removed (Supplementary Fig. 5). Interestingly, most environmental samples contained isolates from a single lineage (45), 11 samples contained isolates from two lineages, and one

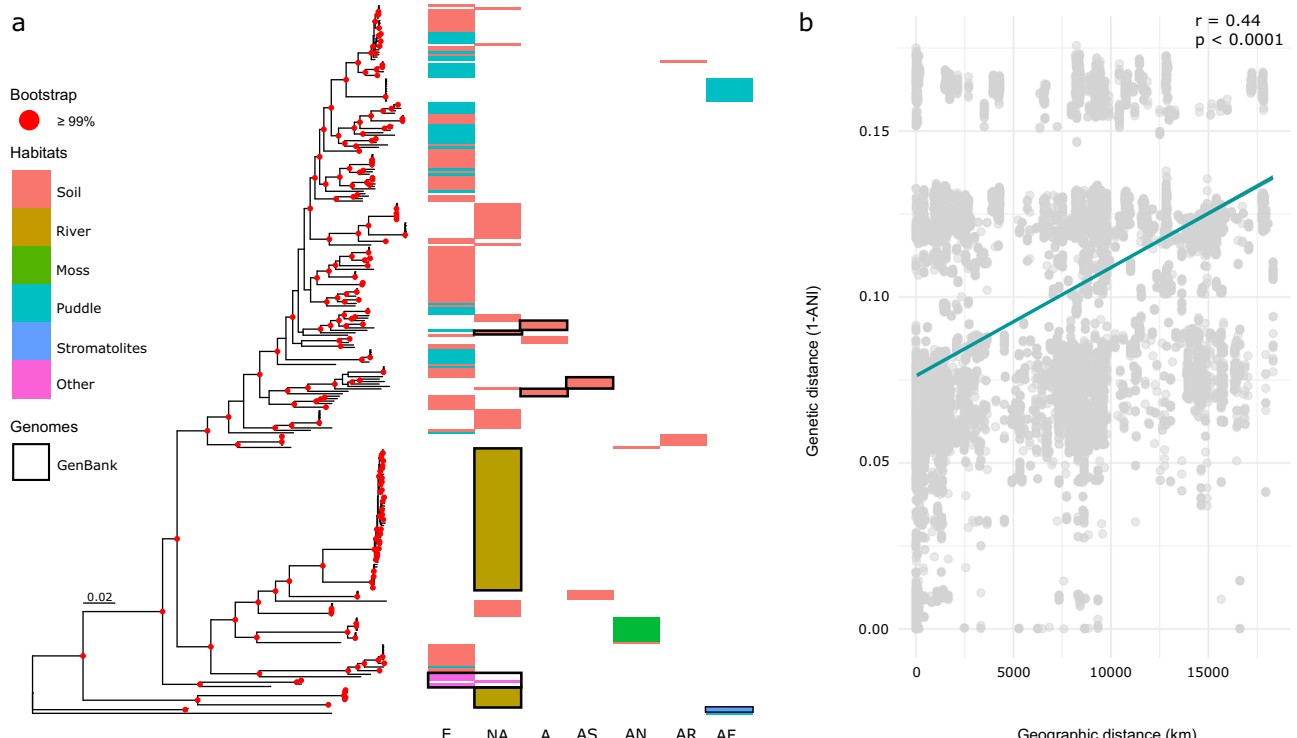

**Fig. 1 | Phylogenomics of the global *Microcoleus* collection and the impact of isolation-by-distance. a** Genome phylogeny based on amino acid sequences of the *Microcoleus* isolates sequenced in this study with the genomes obtained from the GenBank marked in black boxes (dataset II). The strains are annotated with colors corresponding to their habitats. Geographic areas of origin are shown in columns (E Europe, NA North America, A Asia, AS Australia, AN Antarctica, AR Arctic, AF Africa). Nodes colored in red represent bootstrap support ≥99. The scale bar represents substitutions per site. **b** Scatter plot demonstrating a significant effect of geography on *Microcoleus* genetic diversity according to the Mantel test (Pearson's *r* = 0.44, *p* < 0.0001).

sample revealed *Microcoleus* isolates belonging to three distinct lineages (Supplementary Data 3; Supplementary Fig. 3).

Both snapclust and Bayesian optimized clustering analyses recovered 13 clusters as the best-supported partitioning of *Microcoleus*, while unoptimized clustering yielded 18 clusters (Supplementary Data 4). Genome Taxonomy Database (GTDB)[48] classified only three genomes as *Microcoleus asticus*, 19 as *Microcoleus* sp., while the rest (179) were only classified until the genus level (Supplementary Data 4). The isolates did not form groups based on their origin from soil or puddles, nor did they cluster according to specific geographic areas (Fig. 1). There was no observable clustering by GC content or genome size. The pairwise average nucleotide identity (ANI) showed that *Microcoleus* isolates shared 86.94–99.9% sequence identity across their genomes (dataset III; Supplementary Data 5), generating 37 clusters using a 95% threshold (Supplementary Data 4). Given that population genetic analyses are sensitive to population definition and that (1) isolates did not cluster according to habitat, geography, and genomic features, (2) the Bayesian optimized method generated the optimal number of clusters following monophyly in the species trees and (3) the ANI thresholds do not reflect variable diversification rates of species, and it is often lower than the standard 95% threshold in microbes[49], we favor the relatively conservative assignment of *Microcoleus* into 13 clusters. From hereon, we refer to these as lineages M1-M13 of *Microcoleus*. As one of them (M13) included a single isolate, we removed that from further population genetic analysis.

### Diversification of *Microcoleus* lineages started during the Eocene/Oligocene Periods

The dating analysis of the 16S rRNA in BEAST (Bayesian Evolutionary Analysis Sampling Trees)[50] was calibrated at an evolutionary rate of 0.001861 substitutions per site per million years and 95% highest probability density (0.000643–0.003079)[40]. According to the dating,

the divergence of our *Microcoleus* isolates commenced between the Eocene and Oligocene, prior to ca. 29.6 Mya (Fig. 2a). The diversification of *Microcoleus* resulted in at least 12 lineages emerging during the middle Miocene to Pliocene Periods, ca. 4.7–13.7 Mya when the climate was generally warmer and wetter than current (Fig. 2a).

### Demographic changes impacted the diversification of *Microcoleus*

For neutrality statistics, Tajima's D values were significantly departing from 0 in all lineages, apart from M1 (t-test, *p* < 0.001, *p* = 0.47). Mean Tajima's D values over the whole genome varied from −1.35 to 1.73, while Fu's F ranged from −1.25 to 1.91 (Supplementary Data 6, Supplementary Fig. 6). Statistically positive Tajima's D values were observed in four lineages (M3, M4, M7, and M10), which suggests that they might be under the balancing selection with signatures of population shrinkage. In contrast, all other lineages exhibited statistically negative Tajima's D values, which implies a recent population expansion and possible influence of the purifying selection on their genetic diversity. Similar patterns were observed with Fu's F (Supplementary Data 6, Supplementary Fig. 6).

### Considerable flexible genome contributes to the diversification of *Microcoleus*

We analyzed the pangenome of *Microcoleus* using the Roary pipeline[51], which grouped the coding DNA sequences into the core and flexible genome, encompassing soft-core, shell, and cloud genomes. The core genome consisted of genes present in all 201 strains, while the soft-core genes were present in 190–200 strains. The shell contained genes present in 30–190 strains, and the cloud had genes present in only 1–30 strains. Overall, the pangenome contained 133,737 genes, with the core and soft-core genomes accounting for 0.54% of the pangenome and having 639 and 87 genes, respectively (Supplementary Fig. 7).

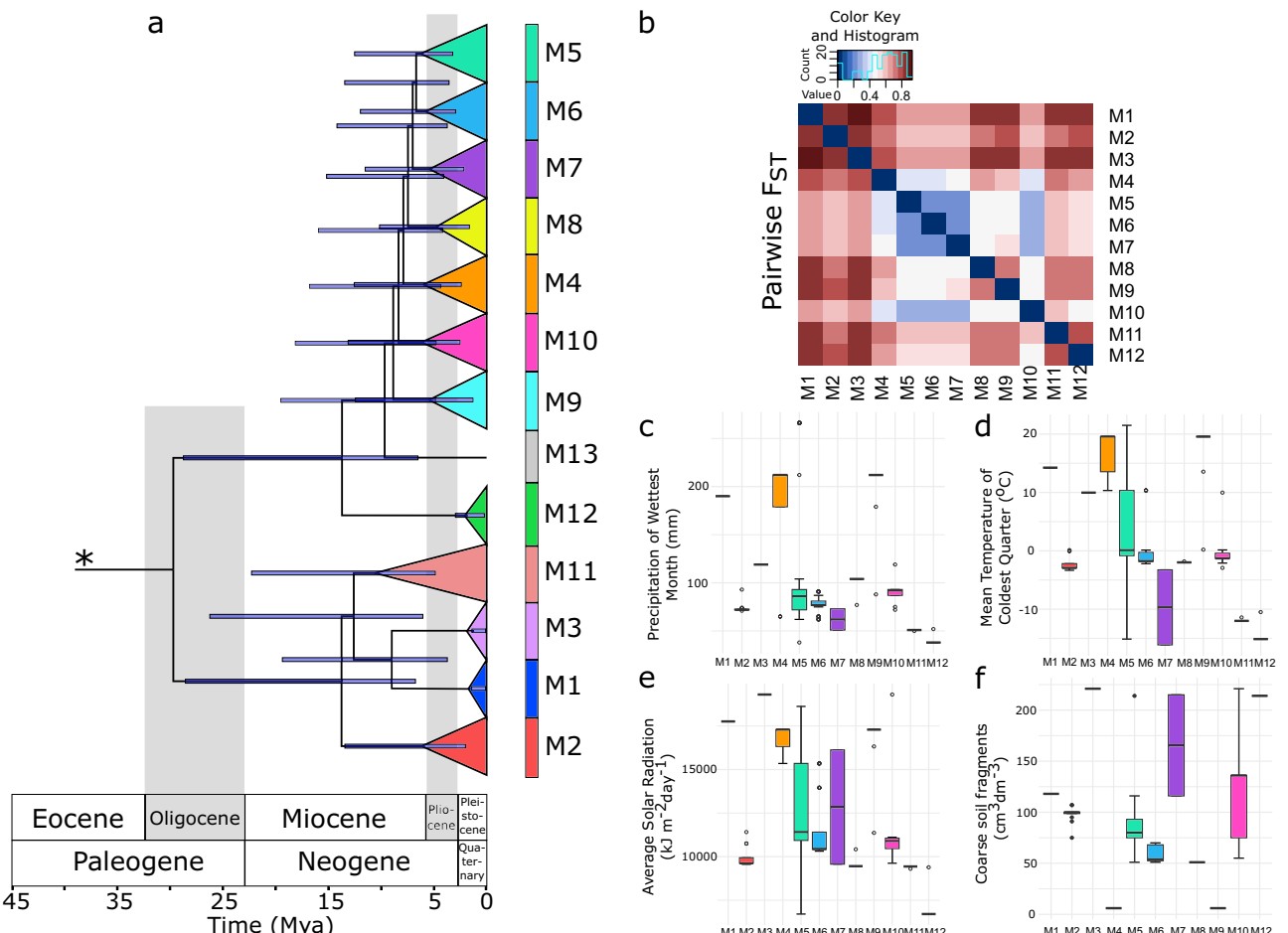

**Fig. 2 | Phylogenetic relationships and divergence times of the 13 *Microcoleus* lineages (indicated as M1-M13) along with genetic and ecological differentiation patterns among them. a** The dating of the divergence times among *Microcoleus* lineages is based on the 16S rRNA gene. Branches were collapsed, and the clades are color-coded according to the lineage designation. The blue bars represent 95% highest posterior density (HPD) values for each node. An asterisk denotes a node of the removed outgroup. The time axis is millions of years ago (MYA) with chronological dating of geological intervals. **b** Heatmap for pairwise $F_{ST}$ distances between lineages. The color spectrum from blue to red illustrates genetic differentiation from low to high. **c–f** Differences in habitat preferences of lineages ($n = 200$ strains) for **c** precipitation of wettest month, **d** mean temperature of coldest quarter, **e** average solar radiation, and **f** volume of coarse soil fragments. Boxplots show the median, interquartile range, default whiskers, and outliers as points. Boxplot colors correspond to the lineages' color codes.

The shell part of the flexible genome comprised 6053 genes, representing 4.52% of the pangenome, while the majority of the pangenome (94.9%) belonged to the cloud part of the flexible genome, which contained 126,958 genes. Further, we used Coinfinder[52] to search for statistically significant associations and disassociations of genes within the shell part of the flexible genome and found the non-random cooccurrence of thousands of genes. In particular, 1,079 genes showed significant associations, while 1,258 were significantly disassociating, suggesting the potential influence of selection on the flexible genome (Supplementary Data 7). The genes were involved in a range of functions, including transport, stress response, cell division, biosynthesis of secondary metabolites, toxin-antitoxin activity, and antiviral defense.

We inferred HGT events in cyanobacterial genomes using a diamond-based HGT detection approach to assess whether HGT has also played an important role in adaptation to novel environments[53]. This approach detected 150,883 gene transfer events in 201 genomes. Of these genes, most could have a cyanobacterial origin (Cyanophyceae and Bacteria in general), followed by other Terrabacteria and orders Nostocales and Leptolyngbyales (Fig. 3a; Supplementary Data 8). Since genome size could affect the HGT rate[54], we performed a correlation analysis between genome size and HGT frequency, and a strong positive correlation was observed (Spearman's rho 0.82,

$p < 0.0001$; Fig. 3b). Thus, we accounted for the genome size by normalizing a number of HGTs per Mb of the genome and compared the frequency of HGT across all *Microcoleus* lineages (Supplementary Data 9), and significant differences were found between them with Dunn's test[55], with the most notable differences between M3, M5, M6, and M12 (Fig. 3c, Supplementary Data 10). In addition, the Kruskal-Wallis test[56] showed significant differences in the HGT rates per Mb between strains occupying different habitats ($p < 0.0001$), suggesting the influence of habitat preference on the HGT levels (Fig. 3d). These results suggest that the variation in HGT rate of *Microcoleus* lineages can be explained by environmental factors rather than the differences in their genome sizes.

Using overlapping 50 kb sliding windows spanning the genome, we evaluated the genetic diversity and divergence for all *Microcoleus* lineages. Estimated mean nucleotide diversities (π) of species over the whole genome ranged from 0.000772 to 0.01067 (Supplementary Data 11). The Kruskal-Wallis rank sum test suggested that π was significantly different between any two lineages ($p < 0.0001$). The mean absolute divergence at the interspecific level spanned 0.011–0.017, whereas the mean fixation index ranged from 0.20 to 0.93 (Supplementary Data 11). The wide range of calculated $F_{ST}$ values, from almost 0 to almost 1, implies that lineage pairs represent distinct stages across the divergence continuum of *Microcoleus*, from early to late stages of

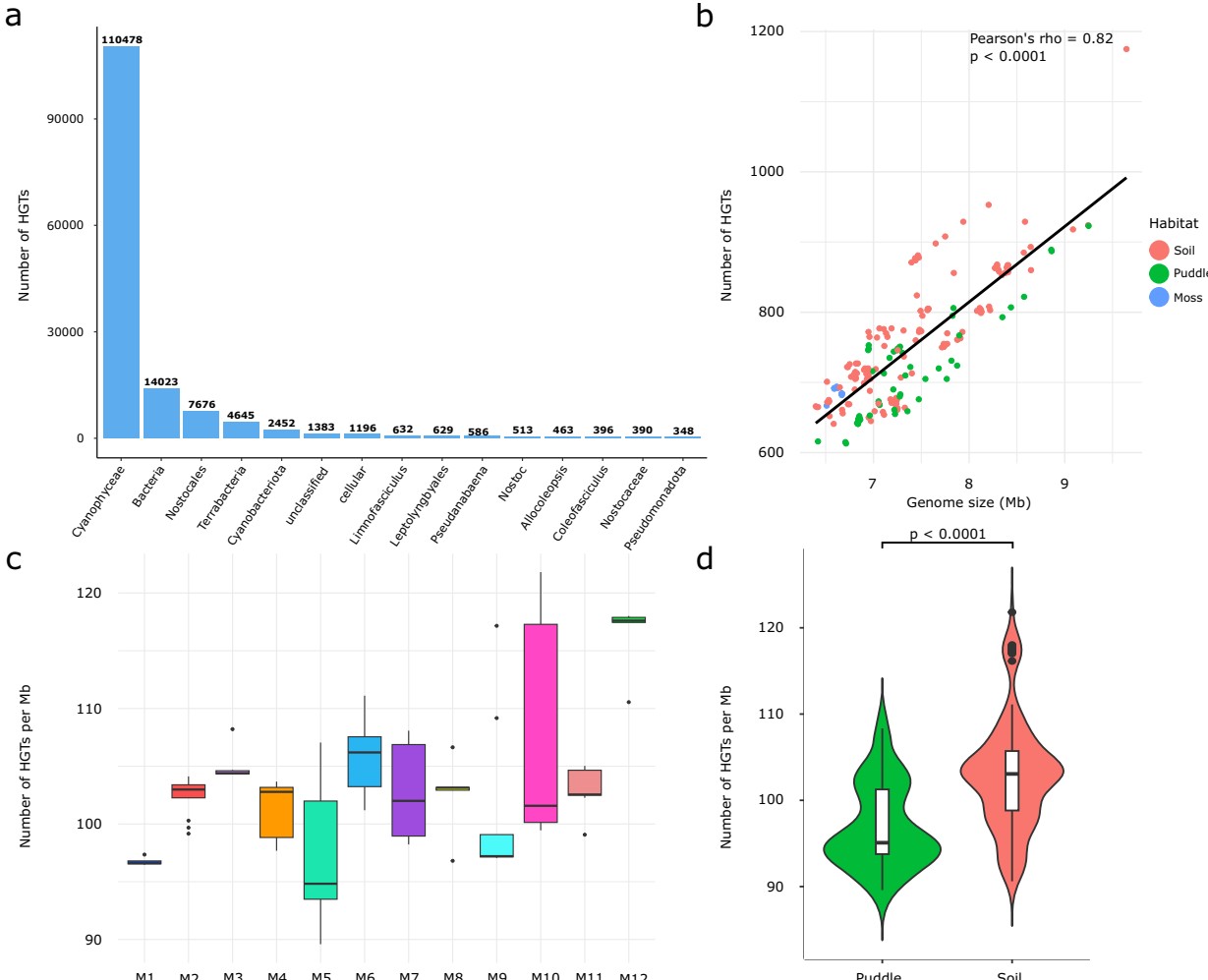

**Fig. 3 | Gene dynamics in *Microcoleus* genomes. a** Bar plot representing the number of detected HGT genes categorized by the suspected source taxa of the horizontal transfer. Only the top 15 most abundant taxa are shown. **b** The scatter-plot representing the one-sided correlation test between the number of detected HGT events and genome sizes (Pearson's $r = 0.82$, $p < 0.0001$). The colors of points correspond to the habitat from which strains have been isolated (red – soil, green – puddle, blue – moss). **c** Boxplots of detected HGT events per genome Mb among *Microcoleus* lineages ($n = 200$ strains). The colors correspond to the lineages' color codes from Fig. 2. **d** The violin plot shows a comparison of the number of HGT events per genome Mb between *Microcoleus* strains ($n = 201$) occupying puddles and soil. Statistically significant Kruskal-Wallis test ($p < 0.0001$) is indicated on the top bar. Boxplots show the median, interquartile range, default whiskers, and outliers as points.

speciation (Fig. 2b). Moreover, high interspecific genome-wide $F_{ST}$ values detected between lineages indicate strong genetic differentiation, with the overall pattern of high heterogeneity along the genome and less evident peaks of elevated divergence in *Microcoleus* genomes (Supplementary Fig. 8).

**Environmental heterogeneity contributes to the diversification of the *Microcoleus* global continuum**

The differences in environmental variables present in the habitats of *Microcoleus* lineages are shown in Fig. 2c–f (Supplementary Data 12 for detailed results of statistically significant pairwise comparisons between environmental variables among lineages estimated by Dunn's test). The lineages M4 and M9 are adapted to habitats with high precipitation and temperature ranges and are significantly different from most other lineages (Supplementary Data 12). Further, lineages M1, M3, M4, and M9 occupy habitats with high average solar radiation, while M3, M7, and M12 favor habitats with a larger volume of coarse fragments than the other lineages (Fig. 2c). Conversely, M7, M11, and M12 prefer dry, cold habitats with low radiation levels. In addition, M4 and M9 opt for soils with finer particles than the other species. Overall, the *Microcoleus*

lineages showed significant differentiation based on their ecological preferences.

To assess the role and identify potential environmental factors that could drive selective pressures in lineages' ecological niches, we performed tests for isolation by the environment (Supplementary Data 13). We employed the phylogenetic signal test[46,47] to examine the similarity pattern of environmental variables considering isolates' evolutionary relationships and utilized the Mantel test to assess the correlation between environmental variables and the genetic distance among the isolates (dataset II). All tested environmental variables showed a statistically significant phylogenetic signal, where Pagel's λ ranged from 0.95 to 0.99 ($p < 0.0001$) and Bloomerg's K had values below 0.001 ($p = 0.001$). These results indicate that the isolates' preferences for environmental niche space align well with their phylogenetic relatedness but might be evolutionarily constrained by other factors. Next, the Mantel test confirmed that contemporary climatic and habitat conditions govern the divergence of *Microcoleus* lineages. The analysis showed that selective pressures were likely triggered by precipitation (annual range, $r = 0.311$, $r = 0.001$; the wettest month, $r = 0.439$, $p = 0.0001$), soil properties (organic carbon, $r = 0.244$, $p = 0.0001$; coarse fragments volume, $r = 0.309$, $p = 0.0001$), and

human activities (net primary production remaining in the ecosystem after harvest, $r = 0.547$, $p = 0.0001$; total human appropriation on net primary production, $r = 0.515$, $p = 0.0001$; nitrogen fertilizer levels ($r = 0.235$, $p = 0.0001$). Overall, these findings suggest that the genomic diversity and population structure of cosmopolitan *Microcoleus* lineages may be shaped by adaptive selection.

## Gene flow delimits *Microcoleus* lineages along the speciation continuum

From the list of recombination events detected by Gubbins[57], we extracted all genomic regions that underwent recombination per strain. The mean genome fraction subjected to gene flow between strains of the same lineage varied from 15 to 53.2% (within the lineages), while the genome fraction affected by gene flow from outside the lineages varied from 1.54 to 29.04% (Fig. 4a, b; Supplementary Data 14 and 15). Genome fractions resistant to gene flow between the evolving lineages have been used as a proxy of the divergence

probability to define lineages according to the UPCEL[21] (Supplementary Data 16). The mean ρ/θ (ratio of recombination relative to mutation rate) and r/m (ratio of the number of SNPs introduced through recombination relative to mutation) values estimated per lineage ranged from 0.01–0.04 to 0.38–2.01 (Supplementary Data 17). The distribution of ρ/θ and r/m estimates of isolates in each lineage is shown in Fig. 4c, d. The Kruskal-Wallis rank sum test indicated that ρ/θ significantly differed between some lineages ($p = 0.0002967$), suggesting those must evolve differently regarding their recombination rates. The post hoc pairwise statistical testing using Dunn's test revealed that the recombination rates were significantly different between M4 and M10 (p adj. = 0.0067) as well as M5 and M10 (p adj. = 0.0182). This is indicative that recombination events happen more frequently in lineages M4 and M5 than in M10. Conversely, although the Kruskal-Wallis test indicated that r/m significantly differed between some lineages ($p = 0.0082$), Dunn's post hoc test showed no pairwise differences. These results imply that selection

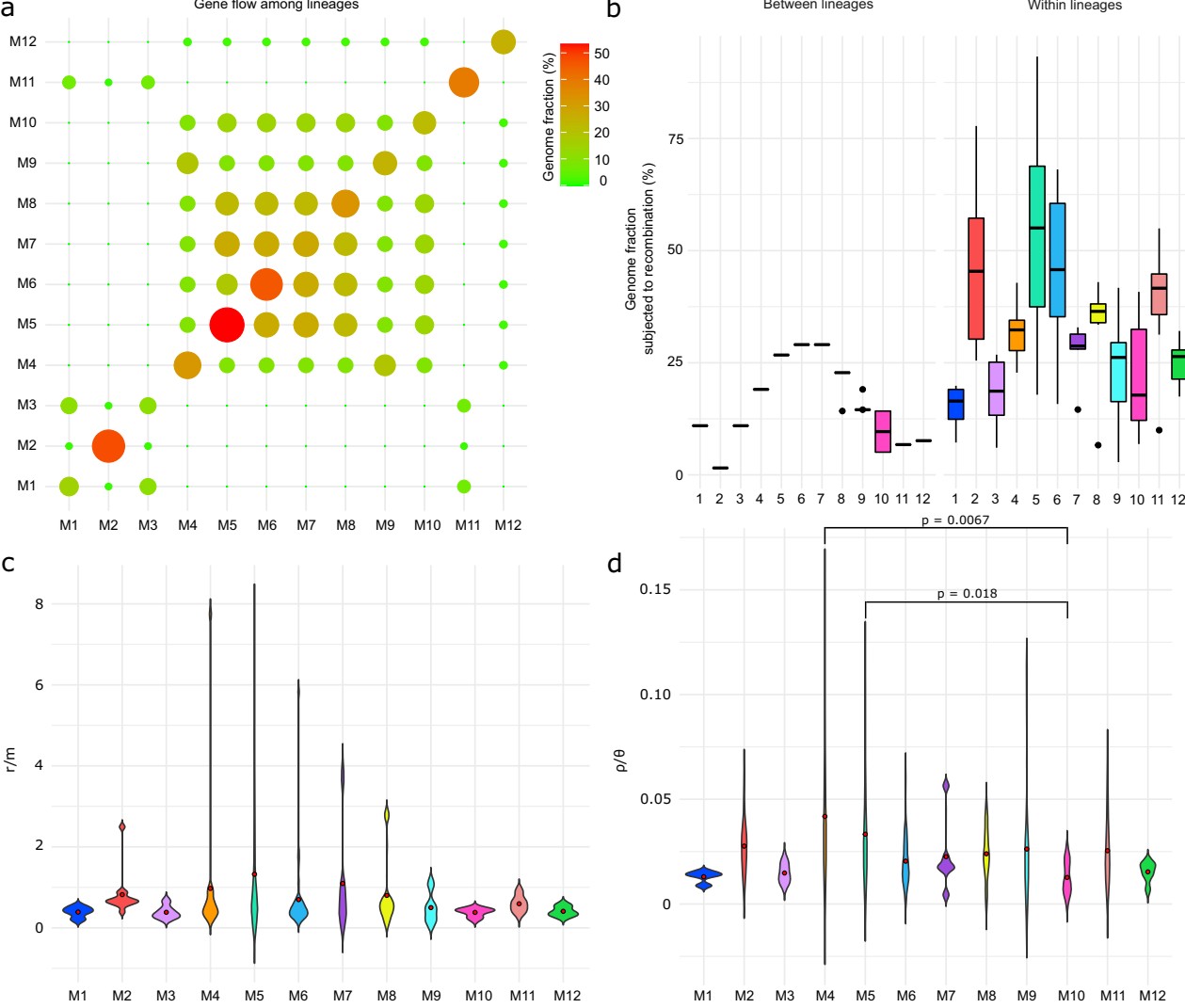

**Fig. 4 | Patterns of gene flow within and between lineages along with violin plots of the per-strain recombination parameters. a** Bubble plot illustrating the extent of gene flow among lineages based on the genome fraction subjected to recombination shared between a lineage pair. Each bubble's size and color correspond to the gene flow level, where the absence and limited gene flow between lineages are shown in green, while high levels are in red. **b** Boxplots indicating the precise variation of genome fraction subjected to recombination between and within lineages ($n = 200$ strains). Boxplots show the median, interquartile range,

default whiskers, and outliers as points. **c** Violin plots illustrate differences in r/m ratios for each lineage. The highest outlier for lineage M5 (r/m = 18) was removed for clarity ($n = 199$). **d** Violin plots of per-strain ρ/θ values for each lineage ($n = 200$). Significant differences between lineages are indicated as p-values, determined by Kruskal-Wallis and Dunn's test with Bonferroni correction for multiple testing ($p_{M4-M10} = 0.0067$; $p_{M5-M10} = 0.018$). The colors of violin plots correspond to the lineages' color codes from Fig. 2. The red dot represents the average value of parameters.

might not have yet had a chance to remove deleterious mutations introduced by recombination owing to lineages being closely related and recently diverged. In addition, the Kruskal-Wallis test showed significant differences in the recombination parameters between strains occupying different habitats (ρ/θ, p = 0.001; r/m, p = 0.0009), suggesting the influence of habitat preference on the variability in HR levels (Supplementary Fig. 9a, b).

## Genes inherent in stress response and biosynthesis of secondary metabolites exhibit signatures of positive selection

Numerous genes are involved in genetic divergence and ecological differentiation among regions of elevated $F_{ST}$ and $D_{XY}$ values (99th percentile). We found 58 annotated genes and 74 genes with a hypothetical function in four genomic regions (highlighted orange in Supplementary Data 11). A total of 19 genes were found to be under positive selection (14 annotated; neutrality index - NI < 1), 13 were found to be under negative selection (10 annotated; NI > 1), seven under both negative and positive selection (5 annotated) in different lineages, and the rest did not show the signature of selection (Table 1). Annotated genes with positive selection had functions associated with stress response (10), chemotaxis (1), biosynthesis of secondary metabolites (6), cell wall formation (1), and transposition (2). Despite not being in the region of high genetic differentiation, other genes may play important roles in the divergence of lineages as the selection was widespread across the genome (Supplementary Fig. 10).

## Discussion

The speciation continuum can range from the emergence of adaptive variation within individuals that freely exchange genes in a population to complete barriers to gene flow between genetically and ecologically distinct species[3,19,58]. A scenario with prevalent population-specific HR (i.e., dominant gene flow within the lineages) conforms to the biological species concept[59]. However, the biological species concept requires fully developed barriers to the gene flow between two species. Some *Microcoleus* lineages have completed barriers to gene flow, while others remain in an intermediate, grey zone (Fig. 4a, b). As a result, assigning lineages to distinct units is not feasible until they have fully diverged as required by biological and any other species concepts[21]. A new approach to studying currently diverging species is necessary, and a universal, probabilistic concept of evolutionary lineages has been developed for this purpose: UPCEL[21]. UPCEL places probabilities on the lineages based on their divergence through gene flow, which can vary over time. In other words, putative species are not defined discretely, but their divergence is described using the probability at a given time.

Following the speciation frameworks of Shapiro & Polz[19] and Kollár et al.[21] and the concept of the speciation continuum of Stankowski & Ravinet[8] adapted to prokaryotes, we considered multiple patterns of genetic differentiation. We observed that $F_{ST}$ roughly coincides with the extent of gene flow (Fig. 2b, Supplementary Fig. 8) and HR (Fig. 4; Supplementary Data 15), where some *Microcoleus* lineages exhibited much genetic differentiation, while some retained the signature of admixture with others, thereby, illustrating *Microcoleus* gradual divergence even with gene flow. We captured all the features necessary to characterize some of the *Microcoleus* lineages as species and place them along the continuum, including preferential gene flow within the groups and variations in the extent of geographic, ecological, and genetic barriers impeding gene flow[8,58]. Hence, we estimate that M1, M2, M3, M11, and M12 are on separate evolutionary trajectories, and they can be considered separate species based on the biological species concept (stages 4–5)[19]. Moreover, regarding UPCEL, they have a > 93.24% probability of becoming fully separated species (between M1 and M3 is 89%; Supplementary Data 16). On the other hand, lineages M4, M9, and M10 remain partially isolated and in the grey zone (stages 2–3) with a probability of >85.7% of eventually reaching speciation completion. The remaining M5, M6, M7, and M8

have a probability of >73.2% becoming fully separated species, and they are likely in the early speciation stages (stages 1–2); thus, they cannot be considered different species yet.

Earlier studies[13,14,60] focused on a handful of incipient species comparisons and noted the presence of genomic regions housing genes associated with adaptation to local ecological microniches. In contrast, we observed a continuum of 12 *Microcoleus* lineages at different stages of genetic divergence (Fig. 2a, b). We found no apparent peaks of elevated differentiation (neither 'islands' nor 'continents'; *sensu* Shapiro & Polz[19]) across the genome but rather a broad elevation of its diversity (Supplementary Fig. 8) as previously observed in some prokaryotes, such as, *Laspinema*[16] and *Sulfolobus*[14] and eukaryotes, such as *Heliconius*[26] or *Rhagoletis*[61]. Broad elevation of genome-wide diversity coupled with selection acting on numerous loci scattered across the genome (Supplementary Fig. 10) and the pronounced effect of HR (Fig. 4) and HGT (Fig. 3) potentially contribute to the ability of *Microcoleus* to respond quickly to sudden environmental stimuli. Given the heterogeneity of soil systems and their selective pressures on organisms, having multiple genomic regions under selection may bring evolutionary advantages compared to having small genomic regions ('islands') with few specific genes. On the other hand, the divergence patterns and 'islands' in *Microcoleus* may exist but have only a temporary character after the initial split of the lineages; hence, they may have been overlooked by the genome scans.

The high ratios of r/m (mean 1.10–2.01; Fig. 4c; Supplementary Data 17) for lineages M4, M5, and M7 indicate that HR has been important for the introduction of SNPs in *Microcoleus* more than mutation, even though low ρ/θ ratios (mean 0.02–0.04; Fig. 4d) suggest that mutation has occurred more frequently. In contrast, the rest of the lineages had lower r/m values (mean 0.38–0.82), suggesting mutation might play more significant roles in introducing polymorphisms within these *Microcoleus* lineages. These estimates aligned with previous reports of recombination rates for cyanobacteria and free-living bacteria[10]. Six of the 12 lineages studied had one to a few highly recombinant genotypes (n = 23, r/m > 2, Supplementary Data 17) potentially responsible for acquiring novel adaptive alleles from the environment. The ability of some *Microcoleus* individuals to fine-tune their mutation and recombination rates, also called fitness-associated recombination[62], might provide them an advantage in local microniches. Individual strains with low-fitness genotypes might have genetic modifiers that increase recombination rates, which would be favored by selection when exposed to abrupt changes in environmental conditions like desiccation or nutrient limitation[62,63]. However, the number of recombinant genotypes obtained from other *Microcoleus* lineages was limited. They might occur in nature but remain unsampled, so the mechanisms driving the diversification may become more complex and intertwined.

Recombination analyses further suggest the connectivity by gene flow among strains from different lineages despite being separated by large spatial scales, which concurs with observations that *Microcoleus* commonly undergoes gene flow[64]. Bearing a significant role in the diversification of *Microcoleus*, gene flow may introduce new, locally beneficial alleles, enabling adaptation to diverse conditions. Additionally, the genome fraction shared within lineages is much higher than that shared between them, implying the existence of a barrier to gene flow that may maintain strains in cohesive genetic groups (Fig. 4a, b).

Barriers to gene flow might also arise due to sequence divergence between species[65], geographical isolation, or the arrangement of ecological microniches[7]. Microbes have long been thought to have unlimited dispersal based on morphology, summarized by the famous tenet "everything is everywhere, but the environment selects"[66,67]. Thus, microbial species were thought to evolve in sympatry. However, our results concur with previous large-scale studies on the cyanobacteria *Microcoleus*[40,44] and *Raphidiopsis*[68] that found patterns of

**Table 1 | Genes within the 99th percentile of elevated $F_{ST}$ and $D_{XY}$ genomic regions with a significant signature of selection according to the McDonald-Kreitman (MK) test**

| Gene ID (NCBI) | Gene abbreviations (prokka) | Gene name (prokka) | Localization (Number of genomes in which is present) | COG function | Neutrality Index (NI) |
|---|---|---|---|---|---|
| 945060 | dxs | 1-deoxy-D-xylulose-5-phosphate synthase | Core genome (201) | Coenzyme transport and metabolism, Lipid transport and metabolism (COG1154) | * |
| 945064 | hepT | Heptaprenyl diphosphate synthase component 2 | Core genome (201) | Isoprenoid biosynthesis; Coenzyme transport and metabolism (COG0142) | $N < 1$ |
| 33366736 | srrA | Transcriptional regulatory protein | Core genome (201) | DNA-binding response regulator, NarL/FixJ family (COG2197) | $N < 1$ |
| 946396 | cheR | Chemotaxis protein methyltransferase | Flexible genome (149) | Signal transduction mechanisms (COG1352) | * |
| 946607 | COQ3_4 | O-methyltransferase | Flexible genome (49) | Ubiquinone biosynthesis; Coenzyme transport and metabolism (COG2227) | $N > 1$ |
| 948080 | glyS | Glycine-tRNA ligase subunit β | Flexible genome (178) | Translation, ribosomal structure, and biogenesis (COG0751) | $N < 1$ |
| 945752 | hcnC | Hydrogen cyanide synthase | Flexible genome (142) | Aminoacid transport and metabolism (COG0665) | $N > 1$ |
| 57664949 | hemY | Protoporphyrinogen oxidase | Flexible genome (78) | Coenzyme transport and metabolism; Heme biosynthesis (COG1232) | $N > 1$ |
| 948805 | ISAcma16 | IS4 family transposase ISAcma16 | Flexible genome | Mobilome: prophages, transposons (COG3385) | $N < 1$ |
| \ | ISWen2 | IS110 family transposase ISWen2 | Flexible genome | Mobilome: prophages, transposons (COG3547) | $N < 1$ |
| 946720 | menD | 2-succinyl-5-enolpyruvyl-6-hydroxy-3-cyclohexene-1-carboxylate synthase | Flexible genome (20) | Coenzyme transport and metabolism; Menaquinone biosynthesis (COG1165) | $N < 1$ |
| 945065 | panE | 2-dehydropantoate 2-reductase | Flexible genome (163) | Coenzyme transport and metabolism; Pantothenate and Coenzyme A synthesis (COG1893) | * |
| 946712 | pchA | Salicylate biosynthesis isochorismate synthase | Flexible genome (77) | Coenzyme transport and metabolism; Menaquinone biosynthesis (COG1169) | $N > 1$ |
| \ | phrB | (6-4) Photolyase | Flexible genome (95) | Uncharacterized protein related to deoxyribodipyrimidine photolyase (COG3046) | $N > 1$ |
| 852135 | pknD_39 | Serine/threonine-protein kinase PknD | Flexible genome (16) | Signal transduction mechanisms (COG0515) | * |
| 946209 | ppsA_1 | Phosphoenolpyruvate synthase | Flexible genome (122) | Carbohydrate transport and metabolism; Gluconeogenesis (COG0574) | $N > 1$ |
| 946393 | rcp1_3 | Response regulator rcp1 | Flexible genome (146) | Signal transduction mechanisms; Chemotaxis (COG0784) | $N < 1$ |
| 946611 | rcsC_52 | Sensor histidine kinase RcsC | Flexible genome (37) | Signal transduction mechanisms (COG0642) | $N > 1$ |
| 946611 | rcsC_74 | Sensor histidine kinase RcsC | Flexible genome (35) | Signal transduction mechanisms (COG0642) | $N < 1$ |
| 946611 | rcsC_75 | Sensor histidine kinase RcsC | Flexible genome (132) | Signal transduction mechanisms (COG0642) | $N < 1$ |
| 946611 | rcsC_76 | Sensor histidine kinase RcsC | Flexible genome (50) | Signal transduction mechanisms (COG2202) | $N < 1$ |
| 946611 | rcsC_78 | Sensor histidine kinase RcsC | Flexible genome (12) | Signal transduction mechanisms (COG0642) | $N > 1$ |
| 946611 | rcsC_93 | Sensor histidine kinase RcsC | Flexible genome (33) | Signal transduction mechanisms (COG0642) | $N < 1$ |
| 946611 | sasA_33 | Histidine kinase | Flexible genome (50) | Signal transduction mechanisms (COG2202) | $N > 1$ |
| 946611 | sasA_43 | Adaptive-response sensory-kinase SasA | Flexible genome (77) | Signal transduction mechanisms (COG0517) | $N < 1$ |
| 947913 | sasA_44 | Adaptive-response sensory-kinase SasA | Flexible genome (142) | Signal transduction mechanisms (COG0745) | $N < 1$ |
| — | trmR_2 | tRNA 5-hydroxyuridine methyltransferase | Flexible genome (182) | Translation, ribosomal structure, and biogenesis; tRNA modification (COG2520) | $N > 1$ |
| 944813 | murF | UDP-N-acetylmuramoyl-tripeptide--D-alanyl-D-alanine ligase | Flexible genome (100) | Cell wall/membrane biogenesis; Mureine biosynthesis (COG0770) | * |
| 946915 | hypothetical | Putative oxidoreductase | Flexible genome | Energy production and conversion (COG0243) | $N < 1$ |

Given are their gene IDs (from the NCBI), abbreviations and names (from prokka annotations), localizations, a short functional description (COG), and a neutrality index (NI). The NI < 1 indicates positive selection and NI > 1 indicates negative selection. Loci under positive and negative selection between pairwise lineage comparisons are marked with an asterisk.

genetic differentiation consistent with the isolation by distance (Fig. 1b). Hence, the existence of some geographical barriers on a large scale (i.e., allopatric differentiation) is suggested. Gene flow between *Microcoleus* lineages persisted across spatially distinct sites, such as between North America and Europe and other regions (Supplementary Fig. 2). Overall, these findings suggest that geography can be a contributory factor in the diversification of *Microcoleus*[18,40].

The preference for climatic niche space may have already emerged as the *Microcoleus* lineages tended to occupy niches with significantly different precipitation levels, soil properties, temperature, and light regimes (Fig. 2c–f, Supplementary Data 12), providing

support for isolation by the environment. Varied response strategies to shifts in environmental selection pressures might explain the patterns of variable HR (Fig. 4c) and HGT rates (Fig. 3) observed in some *Microcoleus* individuals. Differential HR and HGT rates could have been generated through temporal and spatial heterogeneities by favoring different beneficial alleles or modulating processes such as adaptations to a new environment[69]. Additionally, strains found in soil often had lower HR but higher HGT levels than those found in puddles, which could imply that soil environments are more fluctuating (Fig. 3d; Supplementary Fig. 9). However, more data is needed to draw more robust conclusions. These observations underlie a different ability to

respond to stress and nutrients between lineages, which could also contribute to ecological differentiation.

Differences in a huge accessory genome (94.9% of the pangenome), a significant influence of environmental factors on the HGT rates, and a non-random association of hundreds of genes might have further contributed to fine variations in fitness between *Microcoleus* lineages[70]. Nevertheless, these results should be taken with caution as Roary is known to overestimate the size of the accessory genome due to gene fragmentation into different clusters[71]. Interestingly, in the regions of elevated differentiation (outliers in the 99th percentile of $F_{ST}$ and $D_{XY}$), the majority of positively selected genes were part of the flexible genome. The MK test revealed that positively selected genes were functionally linked to stress response (Table 1), a characteristic that has been observed during the diversification of cyanobacteria (*Prochlorococcus*[31]; *Laspinema*[16]) and other prokaryotes (*Vibrio*[13]). The fact that these genes likely initiated the process of divergence means that ecological adaptation or ecological barrier to gene flow has already been established between some of the *Microcoleus* lineages.

Investigating the putative function of the genes found in the region of elevated differentiation may provide an understanding of *Microcoleus'* success. The *rcsC* locus encodes a transmembrane sensor kinase, which plays a critical role in phosphorelay, a signal transduction mechanism associated with various responses to environmental stimuli, such as osmotic shock and capsule synthesis[72]. The RcsC proteins are also known to serve as crucial virulence factors in pathogenic bacteria[73], potentially contributing to host interactions. Further, genes regulating the biosynthesis of secondary metabolites like menaquinone (*menD*, *pchA*) or different cofactors (*panE*) contribute to the ability of bacteria to generate energy for essential processes like resource exploitation or protection against oxidative stresses (Table 1). Indeed, transcriptomic experiments on cyanobacteria showed the same gene function as our selection analysis. The desiccation and rehydration experiments confirmed that many genes linked to signal transduction (e.g., *che* locus), protein transports, and biosynthesis are expressed when exposed to stress[74,75]. These findings indicate that adapting to xerotolerance and osmotic pressures in microniches of arid environments may involve precise genetic adjustments in a few key functional pathways.

The initial radiation of *Microcoleus* lineages started during the transition between the Eocene and Oligocene Periods (before 29.6 Mya, Fig. 2a) when the climate was warmer and wetter than it is today. The split between two major *Microcoleus* clades coincides with the drop in global temperatures[76] and the enhanced aridification period, which might have begun at around 34 Ma and prevailed across the Eocene/Oligocene transition[77]. Molecular dating analysis showed that four *Microcoleus* lineages (M1–M3, M11, M12) diverged around 10–15 Mya in the Miocene, whereas seven (M4–M10) diverged in the late Miocene and Pliocene, around 5–10 Mya (Fig. 2a). However, these analyses stem from the dating analysis based on 16S rRNA, potentially introducing uncertainties regarding the diversification timescale. Compared to macroorganisms that have relatively short divergence times, e.g., mosquitoes, 0.7–4.7 Mya[78]; sunflowers, 1.5–2.2 Mya[79], *Microcoleus* and other microorganisms have much longer divergence times (e.g., diatoms, 0.6–5 Mya[80]; pathogens 20–21 Mya[81]; cyanobacteria, 8–26 Mya[40]; protists, 8–25 Mya[82]). The climate was warm during the middle and late Miocene, with increasing aridification[83]. This trend continued during the Pliocene, which had a cooler climate and experienced high-magnitude shifts in temperature and precipitation[76,83]. Deserts, shrublands, and grasslands also expanded globally[84], possibly providing vacant ecological niches suitable for cyanobacteria to colonize. Consequently, these processes might have enhanced the divergence of *Microcoleus* and facilitated diversification. Furthermore, the ecological and climatic factors might have triggered effective population size expansion or reduction of diversifying

lineages, suggested by departed values of neutrality statistics (Supplementary Fig. 6).

Our study has certain shortcomings. Firstly, particular lineages (M1, M3, M7, M12) analyzed in this study included a relatively small number of isolates (≤6), which may have resulted in an underestimation of their diversity. Additionally, some isolated strains were obtained from the samples collected at one or a few locations in geographic areas (e.g., Africa, Australia, or Asia). More isolates from these lineages should be included in future genomic studies[19]. Secondly, it is possible that other environmental factors, besides those considered in our study, may have also affected the divergence and selection pressures observed between strains. Thirdly, the correlations between variables and genetic diversity presented here do not necessarily imply the direction of a causal relationship (whether ecology and geography drive the barriers to gene flow or vice versa), so further physiological, transcriptomic, taxonomic, and epigenomic explorations will be performed to confirm the role of the putative adaptive genes identified here[85]. Lastly, the clear lack of clear calibration points, especially for free-living (cyano)bacteria, poses a major challenge in dating bacterial evolution, and the use of the 16S rRNA gene likely introduced uncertainties in the estimation of divergence times for *Microcoleus* lineages[86].

The observed diversification patterns of *Microcoleus* presented in this study can be generalized with a model that predicts different stages of microbial speciation, considering various evolutionary forces and mechanisms (Fig. 5). Collectively, our findings suggest that the speciation of microbial species entails divergence over the whole genome by periodic allopatric speciation, ecological speciation with a selection acting on loci in the accessory genome, demographic changes, and extensive gene flow. Further studies on the speciation continuum in other species are needed to reveal how barriers to gene flow emerge and affect microbial speciation and diversification.

## Methods
### Sample collection, whole-genome sequencing, and genome assemblies

Sample collection and culture conditions used followed the procedure described in Stanojković et al.[44] From 57 environmental samples, we selected one to ten strains, extracting the genomic DNA of 202 isolates. Approximately 100 mg of fresh biomass was used, and genomic DNA was isolated with UltraClean Microbial DNA Isolation Kit (MOBIO, Carlsbad, CA, USA). The Nextera XT DNA Library Preparation Kit (Illumina San Diego, CA, USA) was used to mark genome sequence libraries. Isolates were commercially sequenced on the Illumina NovaSeq 6000 platform (Novogene, United Kingdom) at the 150 bp × 2 paired-end mode. Adaptors and low-quality reads were filtered and trimmed using Trimmomatic v0.39[87], and high-quality reads were assembled into genomes using SPAdes v3.13.1[88]. Genome contamination was removed by binning the resulting scaffolds with MaxBin v2.2.4[89], where one bin represented one genome. Genome completion and contamination were assessed via CheckM v1.1.9[90].

Eight *Microcoleus vaginatus* samples collected from North America (3) and Europe (5) between 1851 and 1938 were obtained from the herbarium of the Natural History Museum (London, United Kingdom). Samples were extracted following the Kistler[91] protocol and placed in a sterile Eppendorf tube for subsequent analyses. Following the established protocol by Meyer & Kircher[92], DNA libraries for sequencing were prepared using NEBNext DNA Sample Prep Master Mix Set 2 and Illumina-specific adapters. The libraries were sequenced on an Illumina NovaSeq 6000 instrument (Novogene, United Kingdom) to generate 150 bp paired-end reads per sample. Filtering the reads, assembly, binning, and quality assessment were carried out as previously described.

All the genome sequences were deposited in the GenBank database under the BioProject PRJNA985139.

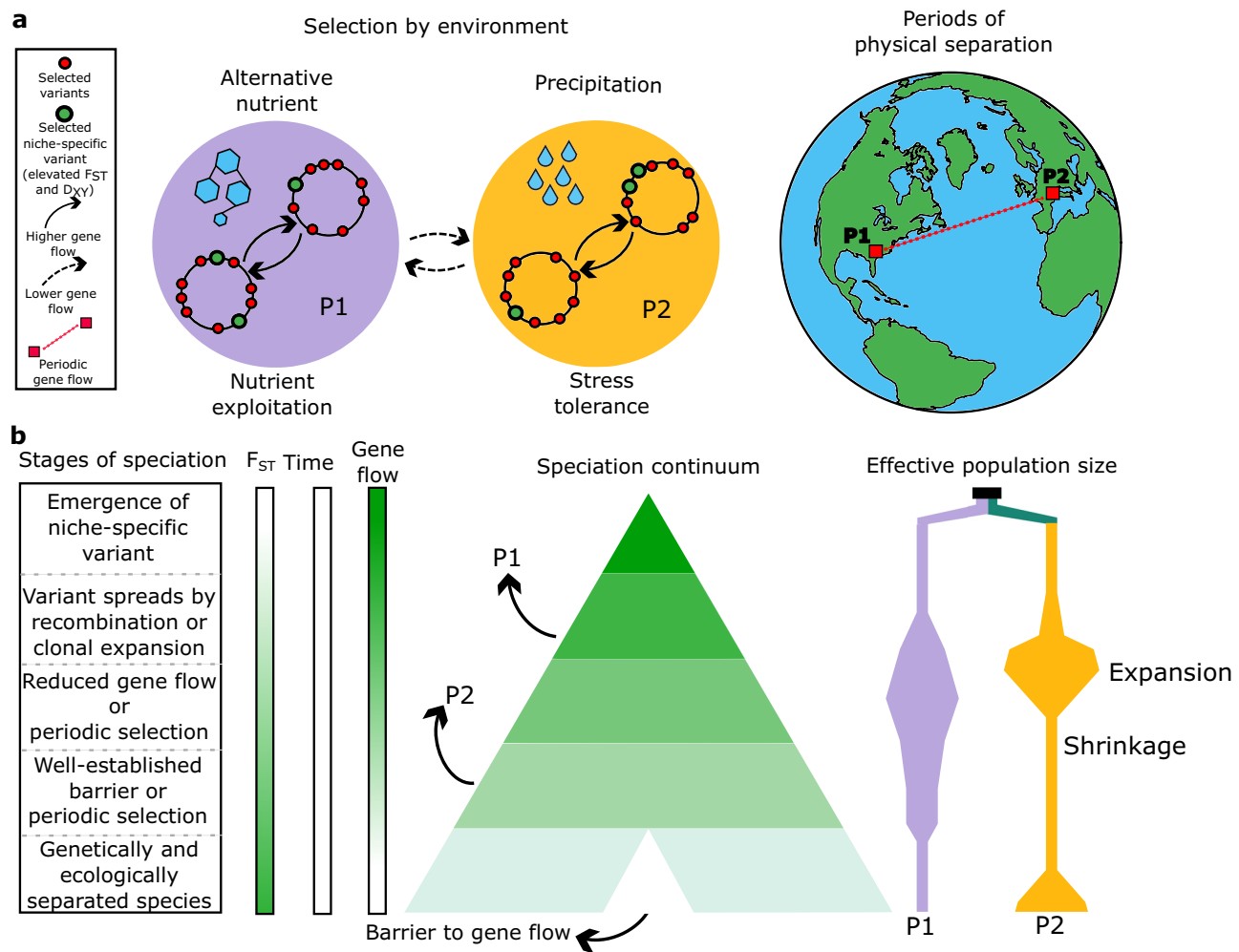

**Fig. 5 | Speciation model in *Microcoleus*. a** Ecological and geographic isolation as drivers of the speciation process. In this example, two *Microcoleus* populations (P1 and P2) have diverged for nutrient exploitation and stress tolerance. Selection acts on variants scattered across the genome (red circles) and variants within the regions of elevated $F_{ST}$ and $D_{XY}$ (green circles), which likely represent niche-specific adaptations. The higher intra-specific gene flow maintains them in cohesive ecological units. Novel ecological niches arose, and periodic physical separation on a large scale interchanges with periods of gene flow. Geographic and ecological barriers to gene flow cause the speciation event. The globe was created with the R package globe v1.2.0[125]. **b** Divergence of two *Microcoleus* populations along the speciation continuum. Populations undergo demographic changes and diverge over time, with different mechanisms acting at different stages of speciation. Barriers to gene flow spread as the genomes become more diverged, eventually resulting in complete ecological and genetic isolation. We can position diverging populations along the speciation continuum by considering various evolutionary forces and mechanisms.

## Genome annotation, variant calling, and filtering

Automated functional annotation for each assembled genome was performed using the prokka v1.14.5 package with default parameters[93]. Filtered and trimmed Illumina sequencing reads were mapped to the reference genome of *Oscillatoria nigro-viridis* PCC 7112 (Accession no. GCA_000317475), and SNPs were detected by freebayes v1.3.2[94] with default parameters. We excluded indels and multiple nucleotide polymorphisms and kept only biallelic variants. Then, we used snippy v4.6 (https://github.com/tseeman/snippy) with default parameters to filter the low-confidence variants.

## Pangenome diversity

We investigated the global genomic diversity of all *Microcoleus* strains sequenced in this study by characterizing the pangenome in Roary v3.13.0[51] with default parameters. As input to Roary, we used genome annotations in gff3 format. For the visualization, we used roary_plots python script (https://github.com/sanger-pathogens/Roary). To determine the gene cooccurrence (associations) and avoidance (dissociations), we computed networks using Coinfinder v1.0.7[52]. We used

a presence/absence matrix as the input with default parameters and a threshold of 0.05 by Bonferroni corrected exact binominal test.

## Horizontal gene transfer analysis

Automated detection of potential horizontal gene transfer (HGT) events was performed using the HGTector v2.0b3[53] with the Diamond all-against-all similarity search on dataset III. The protein database was compiled on a local machine (October 2023), incorporating all protein sequences from the NCBI RefSeq genomes of bacteria, archaea, viruses, fungi, and protozoa (one genome selected per species). Additionally, it included all genomes classified by the NCBI as reference, representative, and type materials. We then assessed the significance of differences in the frequency of HGT among *Microcoleus* lineages by performing a Kruskal-Wallis non-parametric analysis of variance[56], followed by Dunn's test for multiple comparisons[55], along with Bonferroni correction, using the R package dunn.test v1.3.5[95]. We also tested with the Kruskal-Wallis test whether the HGT rates significantly differed between strains occupying different habitats (soil and puddle).

## Phylogenomic analysis

Dataset I was used to reconstruct evolutionary relationships between selected *Microcoleus* isolates and other cyanobacteria. *Gloeobacter violaceus* PCC 7421 (Accession no. GCA_000011385) was used as an outgroup. We used Orthofinder v2.3.1[96] to infer the MSA, which was then used as input for IQ-TREE v1.6.1[97] and the ML reconstruction. The best-fitting model selected by ModelFinder[98] was LG + I + G and branch supports were computed using ultrafast bootstrapping with 2000 replicates.

Orthofinder[96] was used to obtain the MSA for dataset II, and the ML tree was inferred with IQ-TREE[97]. The ModelFinder[98] module was employed to identify JTT + F + I + G4 as the best-fitting model for the phylogenetic reconstruction. The topology was tested with 2000 replicates.

The species tree of *Microcoleus* isolates sequenced in this study (dataset III) was inferred following three different approaches. We used Orthofinder[96] with default parameters to identify single-copy orthologues, infer unrooted gene trees, and acquire MSA. The first species tree was the ML tree inferred in IQ-TREE[97] based on the best model selected by ModelFinder[98] – JTT + F + I + G4 and ultrafast bootstrap with 2000 replicates. *Microcoleus* sp. was used as an outgroup (M2_D5). The second species tree was constructed using unrooted gene trees with a coalescent-based analysis in ASTRAL-III[99,100]. Individual ML gene trees were inferred from the single-copy alignments in IQTREE[97] based on the model identified for each alignment separately by ModelTest[101]. The third species tree was inferred from extracted consensus sequences as fasta files from all SNPs across the genomes detected by snippy and constructed the whole-genome alignment with *Oscillatoria nigro-viridis* PCC 7122 set as the reference (outgroup omitted). Subsequently, it was used for the ML inference in IQ-TREE[97] based on the best model TVM + F + ASC + G4 and 2000 ultrafast bootstrap replications.

## Designating isolates to genetic clusters

We applied four methods to elucidate the population structure of *Microcoleus* isolates (SNPs from dataset III) and assign them to genetically distinct clusters. First, we used a hierarchical Bayesian clustering algorithm implemented in fastBAPS[102]. FastBAPS clusters were determined using functions optimised.symmetric prior (optimized) and baps prior (unoptimized). Additionally, we used the snapclust function[103], which uses the fast distance-based clustering implemented in the package adegenet v2.1.5[104] in R (version 4.1.3; R Core Team, 2021). To identify the optimal number of clusters within the dataset, we used the Akaike Information Criterion (AIC) and the function snapclust.choose.k. Lastly, the pairwise ANI was calculated with fastANI v1.33[105] with a minimum fraction of the genome shared 0.1 and 1.5 kb fragment length.

We used the GTDB-Tk v2.3.2[48] to classify each genome into *Microcoleus* species with the parameter classify_wf. This workflow uses the closest ANI value to locate the user strain into the closest species in the GTDB.

## Inference of time-calibrated phylogeny and biogeographic history

A time-calibrated phylogeny was inferred in BEAST (Bayesian Evolutionary Analysis Sampling Trees) v1.10.4[50] to estimate the age of splits of *Microcoleus* lineages. The analysis was performed on dataset III using the HKY + I + G model as determined by ModelFinder[98]. The 16S rRNA was used due to its universal distribution across microbes, relatively slow evolutionary rates, and its general use for the taxonomic assignment of bacterial species[106]. The dating was performed using a 16S rRNA mutation rate estimated by Dvořák et al.[40] The 16S rRNA sequences were aligned with mafft v7.453[107,108], and the XML file for the BEAST was prepared in BEAUTi[50]. The MCMC chains were run for 10 million generations and sampled every 1000th generation. The

16S rRNA alignment was used only to estimate dating, while the tree topology was fixed to the tree topology based on the amino acid MSA of dataset III. The tree was dated using strict molecular clocks. After the MCMC run, the ESS values were evaluated in Tracer v1.7.1[109], and they were all above 200. The TreeAnnotator[110] was used to produce the final tree with a burn-in of 25%.

Ancestral geographical area reconstruction was estimated in RASP v4.3[111] using the ML reconstruction under the Bayesian binary model (BBM) with the fixed state frequency model (Jukes-Cantor) and nucleotide rate variation for 50,000 generations. As input, we used ML phylogeny inferred from dataset II. We considered seven geographical areas of origin: Europe, North America, Africa, Asia, Australia, the Antarctic, and the Arctic. We allowed a single lineage could occupy a maximum of three geographic areas.

## Phylogeography, phylogenetic signals, and Mantel tests

We investigated the relationship between genetic distance, geography, and different environmental variables using the Mantel test[45] to explore drivers of genetic diversity and diversification in *Microcoleus*. The distance matrix (from dataset II) was constructed from the ANI values as 1-ANI and used as an input for obtaining mantel statistics. Bioclimatic variables were downloaded from the WorldClim v2.1 database[112] at 2.5 arc minutes resolution and extracted in R. Soil variables were downloaded from the ISRIC SoilGrids (www.isric.org), global UV-B radiation parameters from the glUV database[113] and the global human appropriation of net primary production (HANPP) parameters from the data published by Haberl et al.[114] and then extracted with QGis v3.22.8 software (www.qgis.org).

The spatial matrix was generated from the geographic coordinates of localities with the function distGEO from the R package geosphere v1.5-18[115], and the environmental matrices were calculated with the Euclidean distance method. The Mantel tests were performed in the vegan v2.5.6[116] package in R with 9999 permutations. The significance of correlations was calculated with Pearson's r. We assessed the climatic niche preferences of the lineages, focusing on the variables exhibiting the highest correlations according to the Mantel test, particularly associated with precipitation, temperature, radiation, and soil properties. We then assessed the significance of differences between climatic niches by performing a Kruskal-Wallis nonparametric analysis of variance[56], followed by Dunn's test for multiple comparisons[55], along with Bonferroni correction, using the R package dunn.test v1.3.5[95].

Additionally, we measured two independent phylogenetic signal indices in R, Pagel's $\lambda$[46] and Blomberg's K[47], using functions fitContinuous (Geiger v2.0.1[117]) and phylosignal (picante v1.8.2[118]), respectively. The significance of lambda values was performed with likelihood ratio tests.

## Recombination analysis

The recent recombination within *Microcoleus* isolates (dataset III) was explored with Gubbins v3.1.3[57], and the whole-genome alignment generated for the SNP phylogeny was used as an input. It was run according to the software manual, using -first-tree-builder rapidnj and -tree-builder raxmlng parameters to build a recombination-free phylogeny with RaxML[119]. The outputs were visualized with Phandango v1.3.0[120]. Estimates r/m and $\rho/\theta$ for each population and individual strains were derived from the Gubbins output. Owing to the many overlaps between recombination blocks, we counted all recombination events (both shared and unique) specific to one *Microcoleus* lineage (i.e., one block could have been counted more than once) to calculate the genome fractions subjected to recombination. Consequently, the sum of genome fractions undergoing gene flow specific to one lineage may be larger than 100%. Calculated genome fractions were visualized as bubble and box plots using the R packages reshape2 v1.4.4[121] and ggplot2 v3.3.5[122]. To test if there were any differences in

recombination rates and the frequency of recombination among the populations, we used the Kruskal-Wallis test on all the estimated r/m and $\rho/\theta$ values. The test was then followed by Dunn's test with Bonferroni correction, performed in the R package dunn.test. We also tested whether the recombination parameters significantly differed between strains occupying different habitats (soil and puddle) with the Kruskal-Wallis test.

### Estimation of demography, genetic diversity, and selection

The demographic changes of each *Microcoleus* lineage were explored by calculating neutrality statistics. Tajima's D and Fu's F neutrality statistics were assessed in overlapping 10 kb sliding windows with a 2.5 kb step using the package PopGenome v2.7.5[123] in R. Significant departure of these statistics from 0 was estimated with a t-test in R.

We used three genetic parameters to estimate the intra- and interpopulation genetic differentiation of *Microcoleus* populations. We calculated nucleotide diversity ($\pi$), the fixation index ($F_{ST}$), and absolute divergence ($D_{XY}$) with the package PopGenome. All statistics were estimated in overlapping 50 kb sliding windows with a 12.5 kb step. The outputs were visualized using the R package ggplot2. Differences between nucleotide diversities per population were estimated with the Kruskal-Wallis test. We excluded population M13 for these analyses, which included a single strain (N3_A4).

Genomic regions of elevated genetic divergence among *Microcoleus* lineages were identified with both $F_{ST}$ and $D_{XY}$. Windows within the 0.99 percentile of the highest $F_{ST}$ and $D_{XY}$ were selected, and only genomic regions identified by both measurements were considered responsible for the differentiation between the populations. Further, to infer the overall presence of positive selection, we used the genome annotation of *Oscillatoria nigroviridis* PCC 7112 as a reference. We performed the MK test on all genes with a Fisher's exact significance test in PopGenome. The MK test compares levels of polymorphism and divergence to detect deviations from neutrality regarding nonsynonymous substitutions while also controlling for gene-specific mutation rates. By calculating the NI, where an NI greater than 1 indicates an excess of silent divergence and an NI lower than 1 suggests an excess of nonsilent divergence (i.e., positive selection), the MK test is able to estimate whether a gene is under the effect of selection. For all the genes that were found to be significant and had an NI lower than 1, COGs were obtained from the eggNOG-mapper v2.1.12[124]. The genes found within genomic regions of elevated divergence under the effect of positive selection were considered candidates associated with the differentiation between *Microcoleus* lineages.

### Reporting summary

Further information on research design is available in the Nature Portfolio Reporting Summary linked to this article.

## Data availability

Biosample identification and accession numbers are available in Supplementary Data 1. Previously published whole genome sequences are available under the GenBank accession numbers listed in Supplementary Data 1 and 2. Multiple sequence alignments and datasets used and/or analyzed during the study can be found in Supplementary Data 1-17 as well as at https://doi.org/10.6084/m9.figshare.24710961.v1 and https://doi.org/10.5281/zenodo.10677429[126]. The environmental variables were extracted from the following databases: WorldClim (https://www.worldclim.org/), ISRIC SoilGrids (www.isric.org), glUV (https://www.ufz.de/gluv/), and HANPP database (https://sedac.ciesin.columbia.edu/data/collection/hanpp).

## Code availability

All code needed to replicate these results can be found at https://github.com/dvorikus/Microcoleus-population-genomics.

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

## Acknowledgements

We would like to thank Dale A. Casamatta (University of North Florida, USA) for the critical reading of the manuscript, language correction, and helpful comments. The research was supported by the Grant Agency of the Czech Republic (grant no. 19-12994Y and 23-06507S) and the Internal Agency of Palacký University (grant no. IGA PrF-2023-001). This research also received support from the SYNTHESYS Project https:// www.synthesys.info/ (grant no. GB-TAF-4942), which is financed by the European Community Research Infrastructure Action under the FP7 "Capacities" Program.

## Author contributions

A.S., S.S. and P.D. designed the research and analyzed the data. A.S. and S.S. performed laboratory analyses. A.S. and P.D. performed the computational analyses. A.S. led the process of writing. H.J. commented on the manuscript and edited it. All authors contributed to completing the manuscript.

## Competing interests
The authors declare no competing interests.
