## [Peer Review File · Nature Communications]

The global speciation continuum of the cyanobacterium MicrocoleusReviewer #1 (Remarks to the Author):

In this study, Stanojković and colleagues analyzed the diversity of the *Microcoleus* lineage. Overall, the reported patterns of ecology and phylogeography are very interesting since such patterns have not been reported very often in bacteria. The role that these forces play in bacterial evolution remains poorly understood and this study provides evidence that these forces are at play in bacteria. I believe that the manuscript would benefit from focusing more particularly on these results. The authors conducted many advanced bioinformatic analyses on the dataset, but these results are potentially more complex than the interpretations that the authors made from them (for instance, Tajima's D can be affected by other parameters besides those mentioned by the authors). I would recommend the author to be more conservative with their conclusions. I also have several concerns about the methods used by the authors. Here are more specific comments:

Fig1. Please indicate the scale of the branch lengths of the tree. Also, please indicate whether it is an AA or DNA tree.

Line 164: "likely due to ILS", it could also be due to gene flow.

I would be more conservative when interpreting the results of the skyline plots. Many assumptions are likely violated when running these analyses on bacteria (e.g., the assumption of neutrality).

The Discussion is very long and many paragraphs sound more like they should be in the Results section (they often repeat the Results section too).

"Interestingly, in the regions of elevated differentiation (outliers in the 99th percentile of F_{ST} and DXY), the majority of positively selected genes were part of the flexible genome." Have the authors checked whether these accessory genes are true orthologs and if they have been horizontally transferred? Also, see my next comment.

Table 1: what is the % sequence identity of the genes with high F_{ST} and high DXY? Have the authors checked that they are true orthologs? These analyses are very sensitive to the misinference of orthologous genes. Having conducted similar analyses myself, I am well aware that the construction of orthologous gene families needs to be very clean, or the inclusion of paralogs and xenologs will strongly affect the results.

Line 580: "it is possible that other environmental factors, besides those considered in our study, may have also affected the divergence and selection pressures observed between strains." I agree, this is a big concern and I would recommend the authors to be more conservative when inferring the causes of the observed patterns.

Line 682: since the assumption of neutrality is likely violated, using molecular clocks on bacteria is extremely inaccurate (if even meaningful at all). Did the authors also use the core genome alignment to conduct the dating in addition to the 16S? It would be interesting to have both and to compare the results.

Reviewer #2 (Remarks to the Author):

Concept and definition of prokaryotic "species" remain challenging in present-day (micro)biology. In this manuscript, Stanojkovic et al. present an in-depth population-genomic analysis of 291 genomes from *Microcoleus vaginatus* species of a ubiquitous cyanobacterial genus *Microcoleus*. The authors delineate 12 clusters of genomes and thoroughly document gene flow, genomic gene content, selection and environmental conditions that differentiate these clusters. The study nicely highlights the continuum of bacterial diversity even within "species" and evolutionary forces that drive this heterogeneous continuum.

My biggest concern with the paper is not with the analyses conducted and not with the population-

genomic inferences, but with how the authors assign the label "species" to the identified "clusters". The authors initially detect 37 clusters of the analyzed genomes based on ANI, but then cluster them further into 13 clusters, which also appear as clades on the evolutionary tree relating these genomes. Although the division of these genomes into 13 clusters is somewhat arbitrary (different clustering settings produces 18 clusters, for example), the authors immediately start calling the 13 clusters as species. But what species definition and concept did the authors apply? Based on Figure 3A, there is a lot of inter-cluster gene flow. Why not then consider these genomes grouped into 4 "species": M1-M3, M4-M10, M11 and M12, as these would be the clusters with the reduce between-cluster gene flow?

I suggest that throughout the manuscript the authors keep calling these 13 "clusters" as "clusters" rather than "species", and use "diversification" rather than "speciation" when discussing the processes that create genetic diversity within the analyzed group of genomes. This species-agnostic labeling would also fit better with the model of speciation continuum presented in Figure 4B, as it is evident from the presented analyses that some of the 13 clusters do not (yet?) represent the "genetically and ecologically separated species" because of the detected extensive gene flow. The species-agnostic labeling will also remove the confusion of associating some of the 13 newly-labeled "species" with less than 100% speciation probability of becoming "fully separated species".

Reviewer #3 (Remarks to the Author):

The work holds value to Cyanobacteria specialists, and provides detail on the evolutionary, ecological, and genetic circumstances present in this group. The discussions on speciation and gene flow contributes to the conversation around the biological species concept, as it is applied to prokaryotes.

The authors present a thorough study on species and speciation of the terrestrial cyanobacteria *Microcoleus*. *Microcoleus* can be considered a keystone organisms as it acts as a major producer of oxygen as well as acting as a nitrogen fixer. Using genomes they isolated and sequenced from soil samples dating from the 19th century to present in addition to high quality genomes available from public databases, the authors were able to determine that there are at least 12 distinct *Microcoleus* species that are both genetically and geographically distinct from one another but still group monophyletically when compared to other cyanobacteria genomes. More impactfully the authors determined, through several different methods, the time frame and rate of speciation and compared that with the theoretical environmental conditions at the time and found concordance with expected global environmental changes to the expected time ranges of speciation events. Additionally, because this was a whole genome analysis the authors included genes of interest and the level of their positive or negative selection, with genes involving stress response and biosynthesis being the most positively selected, which the authors theorize might be a driver of speciation as the organisms evolve to enter new niches.

The authors outline an almost exhaustive method that others may use to determine speciation within groups of microbes with a high level of confidence and statistical significance. While the authors developed no new programs or statistical analysis, they did utilize a number of difference existing, published, methodologies and used multiple tests for each part of their findings to support their claims. In all, the authors were more rigorous in their analysis than other similar publications and yet do not make any claims that are not thoroughly supported by their data. The methods are clear, if not over-concise, and thus could easily be replicated with a similar data set, or any other data set. The figures are relevant, understandable, and clean. I recommend recommend publication.

Comments, criticisms, and suggestions

line 22 - species diversification might need a qualifier which species, or samples this refers to.

line 38 - species concept should be plural

Somewhere in the Intro and discussion: The fractionation of the scientific community is striking. Key papers that I would have cited in discussing recombination and microbial species are Retchless and Lawrence (<https://www.science.org/doi/abs/10.1126/science.1144876> - they predate the cited Polz paper by a decade), Dykhuisen and Green (<https://pubmed.ncbi.nlm.nih.gov/1938920/>), and Lawrence et al. (<https://pubmed.ncbi.nlm.nih.gov/12446813/>).

line 82 – add “in eukaryotes: to read remains unresolved in eukaryotes. Also, I suspect this may be different for allo- and sympatric speciation.

line 87 – Should probably have a citation

line 103 – Probably should be “may consist of” or “according to our methods/analysis may consist of”

line 109 and elsewhere (e.g., line 728) – most of the data were not available to this reviewer. The accession number in Table S1 do not yet work. Also, the whole genome alignment should be made available in electronic form.

line 110 - publically should read publicly

Figure 1: has no scale bar. Also are enough differences between the highly related strains. Also, this used amino acids, or? This may not be a good choice for closely related OTUs.

line 149 – I am no expert in the Bayesian Binary MCMC (BBM) model, but it seems strange to me that the deepest node has a high probability for area F, whereas a simple parsimony approach would give it about 50% F

line 153 - It seems to me that there are really two groups of divergence, one for distances below 2000 km, the other for the larger distances. It would be worthwhile to repeat the correlation analysis for only the larger distances.

line 162: "ML tree from SNPs" is this from all the SNPs in the genomes, or just the single copy orthologues?

line 164 - “likely due to incomplete lineage sorting” – why not gene transfer?

line 178 – Populations? Species? Groups?

line 197: "Statistically positive Tajima's D values were observed in four species (M3, M4, M7, and M10), which suggests that species might be under the balancing selection with signatures of population shrinkage"
These species are marked with (E) on Fig 2a as "expanding", but the Tajima's D suggested the population is shrinking. However, the Bayesian skyline plots suggest the population events in Fig2a are accurate.

line 217 - Roary, like most other pan genome assembly pipelines, has the tendency to inflate the pan-genome, especially the cloud and shell genomes. This would be worth to comment on in the discussion. see fig 2 in <https://genomebiology.biomedcentral.com/articles/10.1186/s13059-020-02090-4>

Table 1. The designation of location is not clear. The vertical lines in the column appear to suggest 3 groups, but the third one doesn't have a name.

Line 229-230: it may be more informative to say of those genes showing association/disassociation what part of the pan-genome they belong to (soft core/shell/cloud)

line 264 - Reference for phylogenetic signal test is missing.

line 389: "high ratios of r/m (mean 0.38-2.0; Fig. 3C) between species indicate that gene flow has been important for the introduction of SNPs in *Microcoleus* more than mutation"

If the ratio is below 1 does that mean in some cases mutation is more important for SNP introduction. Also, recombination can occur within a genome, i.e., it doesn't necessarily reflect gene flow/HGT.

line 402 - Geographical distance be mentioned and discussed here ? I might have missed this, but a listing of locations for each "species" would be helpful for this discussion.

line 404: "Barriers to gene flow might also arise due to sequence divergence between species, CRISPR repeats," do any of the CRISPR spacers target other *Microcoleus* species? if so the argument in this paragraph would be stronger, and pinpoint the erection of gene flow barriers.

line 436 "requires fully developed barriers" it is a gradient in recombination frequency. And for most prokaryotes a pretty steep one. "Fully diverged" is a figment of the imagination for most bacteria, and apparently not even true for most eukaryotes either.

line 517 - "Ecological differentiation is essential during the early stages of speciation." Only in the presence of gene flow. In case of geographical isolation not.

Figure 4: The coloring in triangle shaped tree remains unclear. Within each group after a barrier to gene flow is erected, gene flow is high.

Collection bias should be mentioned -- many of the older soil samples came from a few locations.

line 630: what were the parameters for prokka annotation? Independent runs of prokka (i.e., not trained on a model) can affect gene calling (especially if the group contains multiple species) and thus inflate the size of the pan-genome.

Reviewer #4 (Remarks to the Author):

The manuscript from Stanojković et al. entitled "The global speciation continuum of the cyanobacterium *Microcoleus*" is a well written manuscript in which authors sequenced 202 isolated strains and 8 herbarium species classified as cyanobacterial *Microcoleus* and defined monophyletic discrete groups as "species" in different stages of speciation in a global continuum of speciation processes framework.

The new data and the approximation is well performed, therefore the main conclusions are valid. Furthermore, the significance of this study is mainly related with a notable sequencing effort of the 202 isolates and 8 herbarium genomes, in which the introduction of the continuum speciation process in a widely distributed cyanobacterial genus are the main noteworthy results for further studies in prokaryotes. This work will be of significance in the topic of microbial speciation. The methodology is well explained, however as this study is a follow up from a previous one, some data should be better presented for readers that are new in this model cyanobacterium.

The analytical approach is consistent, however as much of the analyses depend on the "species" definition made by authors, this definition should be stronger, and this reviewer thinks that some complementary data and approaches could be helpful to sustain the further analyses. It is not understood why authors did not use the GTDB-tk framework to compare their results.

The environmental information used to delineate niche preferences is not found in the manuscript and is cited from a previous article from the same group. Here, appears a question: How did the authors selected these 202 strains from the total set of 495 isolated ones from the previous article?.

Furthermore, the determination of the split divergence ages is widely discussed across the manuscript, but the window sizes are too wide to sustain these affirmations. Also, the collection dates of the herbarium specimens could be useful in the BEAST analyses.

It is understood that the NCBI data is not still publicly available, and that the accession numbers are available as seen in the supplementary information. However, I would kindly ask you to release the genomes soon. The access URLs to the github repository are not available and in some cases the data is cited to the repository when could be shown as supplementary figures.

While most of the manuscript is clear, other parts need improvements. The introduction is well performed and takes into account several evolutionary theories. The results are well explained, but figures need improvements. The discussion is too long, it should be splitted in subsections with clear headings and authors should reduce redundancy regarding the introduction and results. References are well suited.

The expertise of this reviewer is mainly regarding genome sequencing, database submission, comparative genomics and phylogenomic analyses, while the usage of different metrics to determine genetic divergence are more superficially understood. Several suggestions to improve analyses and new analyses are also proposed by the reviewer. The english writting is well understood and the reviewer is not able to further adress corrections in that issue.,

The opinion of this reviewer is that this article is worth of publication in the Nature Communications journal, but it needs some major revisions wich will be outlined below.

Major comments:

Lines 170-178: This paragraph is essential to justify the further analyses. Is there a review or benchmark analysis which compares these different methodologies?. For this reviewer, the explanation of choosing 13 clusters sound like a parsimonomic decision based on the results of different software rather than a justified decision. Is there a bias in the clustering by numbers of genomes (less clusters with singletons)?. While authors presented the values of ANI and established that there are 37 clusters with values over 95 %, is in agreement with the ranges of ANI values reported 86.94-99.9%, and could reflect variable diversification rates (for example comparing the median and standard deviation of intraspecific ANI values for each cluster). This reviewer suggest at this point to further use the GTDB-tk software (Chaumeil et al. 2022, doi: 10.1093/bioinformatics/btac672) to determine in base of this normalized method (RED values) in delineating thresholds for genomic taxonomic classification. Evemn further, if we have a look into the GTDB web page based on the last R214 database version, we can see that the current *Microcoleus* genus in the GTDB contain 106 genomes divided in 24 different species. It is known, and said by the GTDB authors, that this software is not the reference tool for systematic definitions, however if these different species have more consistency with the geographical and environmental distribution, it would be also an option for the delineation. Summarizing, it is recommended to compare in all clustering methods the consistency of the clusters regarding features as geography, environment, genome size, GC content, and any other features to better justify the selection of the final number of "species". Authors, could change the "species" concept and speak about operational monophyletic clusters for example.

Table 1: This table present most of the results regardin selection in genes. However, some field are not well understood. For example, the Localization should be given for any of these genes, understanding that the presence of the gene and the comparison of the multiple sequences helps to determine the neutrality index. Therefore, the Localization (classification) should bve given for all and also the number of genomes in which is present. The Function column could be more useful if it follows a classification system like that for COG or KEGG classification. Gene IDs could be named as Gene abreviation. Gene ID could be easier to identify for a potential reader, if the NCBI protein acesion ID could be given for a representative sequence of the cluster. It is not know by the reviewer if the Table can have colors in the Journal formatting.

Discussion: The discussion is too long and repetitive, it was hard to rerad for this reviewer and

some (not mandatory) minor suggestions were done regarding each paragraph. It is not known if authors can join results and discussion in the results section and leave the two last paragraphs which explain the model and the drawbacks as a general discussion.

Figure 1. The phylogenomic tree should have more preponderance in the figure 1. The phylogenomic tree reconstruction needs to show the support of the different nodes and a scale of the distances. The coloring of the different continents is not so necessary as they are distributed in different columns (this is only a personal comment, as there is redundant information). The color strips occupy too much space. This figure could be used also to show which genomes are new in this study and also information of the environments. Authors can easily manage and add this information in the ItoI webserver. In panel B, authors should briefly explain the Y-axis or how the genetic distance was determined between each pair of genomes and also explain which dataset was used.

The differences in the genome size between the different isolates is noteworthy, however it is not addressed in the manuscript because most of these differences are given mainly in the flexible genome which is a huge part of the genus pangenome. Here also appears something that could have (or not) importance in the differences in the pangenomes and genome sizes which are the mobile genetic elements. As these elements are thought to enhance the adaptability to different or harsh environments, maybe watching the differences could be important for the different species. Authors could use common tools to determine over the assemblies the presence of these elements: geNomad (<https://github.com/apcamargo/genomad>), MOBsuite (<https://github.com/phac-nml/mob-suite>), or Platon (<https://github.com/oschwengers/platon>).

It was also suggested somewhere by this reviewer, that the effect of HGT or terminal paralogous (gene duplication) in the speciation process could be determined to see other influencing factors and to balance the role of them against recombination and mutation. In the Orthofinder results can be obtained the paralogous sequences, and there are several tools like HGTector (<https://github.com/qiyunlab/HGTector>) to determine genes potentially affected by this process.

Minor comments:

Abstract

line 13: This phrase is unclear about what is driven.

*lines 16-17: This phrase is strange in the context of the study, specifically speaking of the *M. vaginatus* species, because in Table S2 there is only one *Microcoleus vaginatus* genome, while the others are unknown species. line 25: Biosynthetic processes is not a correct term, as much part of the metabolism is biosynthetic. Do you refer to secondary metabolism biosynthesis?

Introduction

lines 52-54: This phrase is hard to digest, as in the previous phrase authors speak about the continuum of barriers (similar to accumulation of barriers) and in this phrase about the strength. Furthermore, the cryptic species is a not well defined concept, therefore mentioning here open question which will not be addressed.

lines 54-55: This phrase doesn't have relevance in the context of genetic differences, as probably you refer to the phenotype as the morphology considering the two cited references; while the microbial world is featured by extensive genetic diversity with similar morphologies.

lines 55-57: Please provide references, or this phrase is supported by those on the next phrase?.

lines 57 - 59: Isolated in which terms, geographically or genetically?

Line 61: It is understood that authors don't mention in the manuscript the Horizontal Gene Transfer mechanism as having roles in diversification because when comparing at genus level, the influence of this mechanisms could be smaller than others, however it would be worth mentioning it (and if it's possible to perform some analyses to compare with the influence of recombination).

line 77: Please fix the parenthesis.

lines 87-88: It is known the expansion of freshwater (origins of *Gloeobacter*) and then marine cyanobacteria regarding the GOE, however it is not known for this reviewer if there is evidence of the importance of cyanobacteria in terrestrial systems in those time scales.

Results

lines 121-125: While in Table SI is shown the high degree of completeness and low contamination of the genomes (lower for the herbarium specimens), it would be better to report here the mean and minimum values of those metrics. Also, it is noted the huge difference of the genome size, the lower genome size obtained is for a lower complete genome?.

lines 125-126: The features of the genomes from the NCBI also should be determined, and only genomes with high completeness and quality should be retained for further analyses.

line 128-129: Please provide somewhere which 30 genomes were used in dataset I.

line 136-137: Please explain somewhere why this strain was used as outgroup.

lines 144-146: Please provide the accession numbers for the Genbank genomes in the Figure S1.

For example, why are two genomes as the PCC 6506?. This is also related with the fact that in Table S2, the names of all genomes of the dataset II were changed, however it should be better to show the original names.

lines 144-146: It is not understood which parameter was used to define the M2_D5 genome as sister taxa and not as part of the Microcoleus continuum. The results for the ancestral geographical results could also change if this strain is included inside the Microcoleus, being a potential African origin as outlined in the text. Please justify.

lines 166-168: Please, explain better this sampling grouping for readers not aware of the methodologies used in the previous study.

line 217: Authors used Orthofinder and Roary software over the same dataset, are the results consistent between both?

lines 221-223: do you mean clusters of genes (be careful when you say CDS)?. The number of clusters of genes being in the core/soft-core is too close with the early determinations on cyanobacteria (Simm et al. 2015, doi: 10.3389/fmicb.2015.00219). Is there an effect in these numbers of the low-completeness genomes?.

lines 228-230: what do you mean with metabolic genes and with the "annotated" in parenthesis?. The URL is not available. Authors also could show these results as supplementary tables.

lines 231-233: please be more specific in which genes are you referring to. One of the functions is biosynthesis, please be more specific.

lines 242-246: maybe is in discussion, however are you referring to a potential absence here of speciation islands as defined in introduction? Please authors can make an effort in summarizing the data in supplementary tables or figures.

lines 251-252: Besides the tests, please provide the raw environmental parameters. Which are all the variables measured in the original samples?

lines 255-259: These reported tests are those only for variables with statistical differences or authors did find differences in the other environmental parameters?

line 272: Microcoleus species.

***lines 272-279: These results are good, maybe they should have further protagonism?.

lines 289-290: "estimates of isolated in each species..." this phrase sounds strange.

lines 290-292: This phrase should be fixed as the Kruskal-Wallis test determines if there is any difference between the groups tested, and in this phrase it reads that every group is different from any other.

lines 305-306: Is there any bias given by the level of assembly of the CRISPR regions? This question is regarding the difficulties on assembling these genomic regions. The CRISPR repeats are the same for all the genomes in this dataset?.

lines 313-314: This URL is not available, and it seems related with the Gubbins results rather than with the CRISPR analyses. Here a supplementary table with the numbers of CRISPR repeats and the values of recombination blocks would be helpful for readers. As suggested above, here authors could show the CRISPR sequences, and if the repetition of the different arrays is different across species.

lines 320-321: Authors say that in four regions are concentrated the mentioned genes; however, besides recombination, it is not mentioned or analyzed if the Microcoleus genomes have extensive genome rearrangements (which could reduce the genomic regions common in the dataset) or if they maintain as much as possible the genome synteny.

lines 324-326: please read the commentary about the Table 1 regarding the function assignment. This is related with the topic that biosynthesis is a wide metabolic function, and maybe authors want to focus on secondary metabolite biosynthesis.

lines 326-330: The figure S9 (S11 doesn't exist) need more detail about which genome was used to map the NI values. This figure shows, as well the text is explicit, that the >1 NI values are

across all the genomes, therefore it is not understood how the previously mentioned 4 regions were determined and in which genome. It is recommended to add a supplementary Table instead a hyperlink to the github repository.

Discussion

lines 336-348: This introductory paragraph to the discussion, should be shortened and as it tries to summarize the findings of the study and many of this information will be taken into account later.

lines 338-339: "a continuum of *Microcoleus* species..." the continuum is referred across the manuscript as regarding the speciation process, please clarify.

lines 360-374: this paragraph should be joined in a section specific to the species and continuum speciation process. Authors should shorten or refocus the information of this paragraph that is given in sections above (lines 365-368)

lines 375-385: the main message of this paragraph is that of lines 379-381, authors can shorten the other phrases (and provide a supplementary table with the *Fst* analyses).

lines 387-394: the main message of this paragraph is between lines 387-394, it is suggested to shorten the other phrases.

lines 402-420: this paragraph is too much long and it mixes too much information. It is not understood the role of natural competence in the rates of homologous recombination, and the CRISPR-cas seem not to be barriers to this process; therefore the topics seem to be intermixed but not justified. Maybe, the discussion of the barriers of recombination due to the loss of the specific 20 bp sequences that are recognized by recombinases could be more informative.

lines 423-429: this paragraph could be shortened and joined in a special section regarding the environmental constraints in speciation.

lines 431-445: this paragraph should be joined in the section about the speciation continuum which has been addressed in the beginning of the discussion.

lines 447-459: this paragraph seem to be more related with results, complementing those analyses rather than properly discussion.

lines 461-499: these three paragraphs could be joined in a section in which a table is presented with the split time divergencies of the different "species" and the climatic trends along with their references. Therefore authors could focus in the main discussion as that of lines 485-487.

lines 521-526: the first of these two paragraphs is too descriptive, therefore the main message is lost.

lines 528-575: these paragraphs could be shortened and focused in the ecological drivers of barriers to HR as well specific genes for adaptations to the different niches in which authors detected preferences between different "species".

lines 577-586: please provide some references in other models for these different suggestions.

Materials and methods

lines 603-604: what about the axenicity or uncyanobacterial state of the cultures? could authors provide the checkM parameters obtained of the assemblies before the binning procedure with maxbin?.

lines 608-609: please provide the parameters of quality filtering. Also, there is a word missing before "... quality reads". Did authors used the SPAdes software in default mode, or used the --meta or --careful options?.

lines 610-612: did authors searched for the rRNAs present in the genomes (and unbinned contigs). Sometimes, even from cultured strains these regions are not binned and valuable information is lost when submitting to NCBI.

line 625: authors are encouraged to release the data before the next round of revisions (if applicable)

lines 628-629: while prokka is the gold standard tool to annotate genomes, to further classified the annotated sequences it is recommended to use the eggNOG-mapper (<https://github.com/eggNOGdb/eggNOG-mapper>) or deepNOG tools (<https://github.com/univieCUBE/deepNOG>) to get the COG hierarchical classification of the genes or the Kofam-KOALA web server (<https://www.genome.jp/tools/kofamkoala/>) to annotate under the KEGG classification.

line 630: it is not understood by the reviewer why authors used that genome and not the *Microcoleus vaginatus* genomes (GCA_022701275.1 or GCA_000214075.2).

lines 652-654: authors can join this with the previous paragraph only outlining the differences in

the procedure (or must to cite correctly as in the previous paragraph the software).
line 673: please provide the reference (doi: 10.1093/nar/gkz361).
line 710: which type of distance measurement was determined with the R package(e. g. geodesic, grat-circle, etc)?
lines 745-748: how authors determined the CRISPR repeats (remove the word spacer)?
lines 776-790: these methods were not completely understood by reviewer in terms of inputs and outputs. SApecially, what was done with the CDS in lines 779-781. What do you refer with "samples" in line 781. In lines 784-787 there is not space given for genes with neutral selection, please explain.

Figure S2. Authors should improve the figure legend regarding what do you refer with the bold and red highlighting. Do you mean genomes/strains grouped in different species coming from the same environmental sample?.

Figure 2: Align the panel letter B) with the others.

Figure 3: In panel A) and in supplementary table 9 the given values are the mean inside and between species groups?. The standard deviation should be given in the table (and I suppose that these deviation values are given in figure 3 B).

Reviewer #1 (Remarks to the Author):

*In this study, Stanojković and colleagues analyzed the diversity of the *Microcoleus* lineage. Overall, the reported patterns of ecology and phylogeography are very interesting since such patterns have not been reported very often in bacteria. The role that these forces plays in bacterial evolution remains poorly understood and this study provides evidence that these forces are at play in bacteria. I believe that the manuscript would benefit from focusing more particularly on these results. The authors conducted many advanced bioinformatic analyses on the dataset, but these results are potentially more complex than the interpretations that the authors made from them (for instance, Tajima's *D* can be affected by other parameters besides those mentioned by the authors). I would recommend the author to be more conservative with their conclusions. I also have several concerns about the methods used by the authors. Here are more specific comments:*

We thank the reviewer for the comments and suggestions. We revised the manuscript to be more conservative with our results. Specific changes are mentioned under each point.

Fig1. Please indicate the scale of the branch lengths of the tree. Also, please indicate whether it is an AA or DNA tree.

The scale has been indicated in the revised figure, and the legend indicates that it is an AA tree.

Line 164: "likely due to ILS", it could also be due to gene flow.

We have added gene flow (Line 148).

I would be more conservative when interpreting the results of the skyline plots. Many assumptions are likely violated when running these analyses on bacteria (e.g., the assumption of neutrality).

We have modified the paragraph with interpretations of the demography results in the discussion. We included possible concerns about the skyline plots. We avoided making definite claims (Lines 450-463).

The Discussion is very long and many paragraphs sound more like they should be in the Results section (they often repeat the Results section too).

We have modified and shortened the discussion following suggestions given by all reviewers.

*"Interestingly, in the regions of elevated differentiation (outliers in the 99th percentile of *F_{ST}* and *DXY*), the majority of positively selected genes were part of the flexible genome." Have the authors checked whether these accessory genes are true orthologs and if they have been horizontally transferred? Also, see my next comment.*

*Table 1: what is the % sequence identity of the genes with high *F_{st}* and high *DXY*? Have the*

authors checked that they are true orthologs? These analyzes are very sensitive to the misinference of orthologous genes. Having conducted similar analyses myself, I am well aware that the construction of orthologous gene families needs to be very clean, or the inclusion of paralogs and xenologs will strongly affect the results.

Our investigation of the region of elevated differentiation (99th percentile of F_{ST} and D_{XY}) is based on the genome scans performed over the reference genome (*Oscillatoria nigro-viridis* PCC 7112). Those genomic regions (windows) with the top 1% of the highest both F_{ST} and D_{XY} , were selected, and we identified what genes are in these regions based on the reference genome.

Detected windows of elevated differentiation were found between the following pairwise lineage comparisons (between M2 and M8, M12; between M4 and M6, M9, M10, M11, M12; between M5 and M10; between M6 and M7, M9, M10). In other words, one region of elevated differentiation does not appear across all the lineages. In Supplementary Data 11 we highlighted only genomic regions with elevated values of both F_{ST} and D_{XY} .

Independent of genome scans, we have performed the MK test for selection on all genes found in genomes of strains from each population (genes were mapped onto the reference genome based on the reference genome annotation data). Then, in the selection analysis results, we searched for the genes found in the region of elevated differentiation to see whether they had been under positive or negative selection.

Hence, these analyses are not sensitive or related to orthologous genes. The reason is that we mapped the reads directly to the reference, not searching all the genes from all of the genomes or selecting genomes against each other.

Line 580: "it is possible that other environmental factors, besides those considered in our study, may have also affected the divergence and selection pressures observed between strains." I agree, this is a big concern and I would recommend the authors to be more conservative when inferring the causes of the observed patterns.

We have modified wording in some parts of the discussion to avoid making overly strong claims and be cautious with our interpretations (Lines 393, 450, 451, 457-463).

Line 682: since the assumption of neutrality is likely violated, using molecular clocks on bacteria is extremely inaccurate (if even meaningful at all). Did the authors also use the core genome alignment to conduct the dating in addition to the 16S? It would be interesting to have both and to compare the results.

We agree that there is a lack of convincing calibrating points as well as significant differences among substitution rates within prokaryotes, especially cyanobacteria, making dating analysis extremely challenging. However, 16S rRNA is one of the few genes used for dating in prokaryotes with a known mutation rate. Importantly, we have calibrated the molecular clock based on the previously estimated evolutionary substitution rates from the fossil cyanobacterial DNA sequences of 16S rRNA (see 10.1371/journal.pone.0040153). The mean substitution rate per million years used was 0.001861 (95% CI, 0.000643-0.003079). We would like to keep these analyses as they offer new insight into the emergence of the continuum of *Microcoleus* species

and lineage divergence rates among cyanobacteria. This is the closest we could be in the dating analyses, as fossil records for cyanobacteria are scant.

We have not used core genome alignment to conduct the dating as we would need to know the evolutionary rates of all genes found in the core genome. That analysis is currently not possible to be performed owing to the lack of information on the substitution rates of each gene. However, the backbone tree for the dating analysis was reconstructed based on the core genome alignment.

Reviewer #2 (Remarks to the Author):

*Concept and definition of prokaryotic "species" remain challenging in present-day (micro)biology. In this manuscript, Stanojkovic et al. present an in-depth population-genomic analysis of 291 genomes from *Microcoleus vaginatus* species of a ubiquitous cyanobacterial genus *Microcoleus*. The authors delineate 12 clusters of genomes and thoroughly document gene flow, genomic gene content, selection and environmental conditions that differentiate these clusters. The study nicely highlights the continuum of bacterial diversity even within "species" and evolutionary forces that drive this heterogeneous continuum.*

My biggest concern with the paper is not with the analyses conducted and not with the population-genomic inferences, but with how the authors assign the label "species" to the identified "clusters". The authors initially detect 37 clusters of the analyzed genomes based on ANI, but then cluster them further into 13 clusters, which also appear as clades on the evolutionary tree relating these genomes. Although the division of these genomes into 13 clusters is somewhat arbitrary (different clustering settings produces 18 clusters, for example), the authors immediately start calling the 13 clusters as species. But what species definition and concept did the authors apply? Based on Figure 3A, there is a lot of inter-cluster gene flow. Why not then consider these genomes grouped into 4 "species": M1-M3, M4-M10, M11 and M12, as these would be the clusters with the reduce between-cluster gene flow?

We understand the reviewers' concerns about the designation of isolates to clusters/species. We started explaining in the results section from Line 153. We discussed different numbers of clusters yielded by different clustering algorithms (snapclust – 13; Bayesian optimized clustering analysis – 13; Bayesian unoptimized clustering analysis – 18; the ANI – 37) as well as the reason for not considering them as our designated lineages (arbitrary thresholds that do not bear any evolutionary significance and lack of monophyly). We have also included information on the lack of clustering patterns according to habitat, geography, and some genomic features (GC content and genome size), together with the results from GTDB-Tk. Moreover, designating isolates to clusters/species based on reduced gene flow between them would mean that such "species" are not monophyletic (for instance, M2 is paraphyletic to M1+M3). Considering everything, we have used the monophyletic species concept, which is a widely used concept to delimit cyanobacterial species. It requires that lineages fall into a distinct monophyletic group/clade with a unique feature (autapomorphy), which can be any difference between the two groups/clusters (e.g., genetic, ecological, morphological). Only the Bayesian optimized

clustering analysis yielded monophyletic clades, and we have assigned our isolates more conservatively into 13 lineages.

Although no apomorphies were investigated in this study, we have already performed morphological assessments of our *Microcoleus* strains. Interestingly, morphology concurred with the assignment of *Microcoleus* to these 13 lineages. We have found that they can be delimited based on different morphological traits as well, such as cell width, cell length, and the pointiness of the apex cell (calyptra). However, including all this information would be out of the scope of this study. We are currently preparing a separate paper, which aims to do a taxonomic revision of the genus *Microcoleus*, where phenotype information would be more fitting.

I suggest that throughout the manuscript the authors keep calling these 13 "clusters" as "clusters" rather than "species", and use "diversification" rather than "speciation" when discussing the processes that create genetic diversity within the analyzed group of genomes. This species-agnostic labeling would also fit better with the model of speciation continuum presented in Figure 4B, as it is evident from the presented analyses that some of the 13 clusters do not (yet?) represent the "genetically and ecologically separated species" because of the detected extensive gene flow. The species-agnostic labeling will also remove the confusion of associating some of the 13 newly-labeled "species" with less than 100% speciation probability of becoming "fully separated species".

We agree with the reviewer that we should have been more cautious about calling our clusters "species" immediately. We think it is better to call these clusters "lineages" at first. Here, we understand lineage as an ancestral-descendant series of evolutionary units (genomes/populations/clusters/groups) connected by gene flow. Therefore, to make a more coherent story, we begin talking about these detected clusters as "lineages" throughout the text until the point in the discussion where we offer evidence for calling some of the lineages "species". We have also modified the text throughout the manuscript to use "diversification" rather than "speciation" in some contexts.

Reviewer #3 (Remarks to the Author):

The work holds value to Cyanobacteria specialists, and provides detail on the evolutionary, ecological, and genetic circumstances present in this group. The discussions on speciation and gene flow contributes to the conversation around the biological species concept, as it is applied to prokaryotes.

*The authors present a thorough study on species and speciation of the terrestrial cyanobacteria *Microcoleus*. *Microcoleus* can be considered a keystone organisms as it acts as a major producer of oxygen as well as acting as a nitrogen fixer. Using genomes they isolated and sequenced from soil samples dating from the 19th century to present in addition to high quality genomes available from public databases, the authors were able to determine that there are at least 12*

distinct Microcoleus species that are both genetically and geographically distinct from one another but still group monophyletically when compared to other cyanobacteria genomes. More impactfully the authors determined, through several different methods, the time frame and rate of speciation and compared that with the theoretical environmental conditions at the time and found concordance with expected global environmental changes to the expected time ranges of speciation events. Additionally, because this was a whole genome analysis the authors included genes of interest and the level of their positive or negative selection, with genes involving stress response and biosynthesis being the most positively selected, which the authors theorize might be a driver of speciation as the organisms evolve to enter new niches.

The authors outline an almost exhaustive method that others may use to determine speciation within groups of microbes with a high level of confidence and statistical significance. While the authors developed no new programs or statistical analysis, they did utilize a number of difference existing, published, methodologies and used multiple tests for each part of their findings to support their claims. In all, the authors were more rigorous in their analysis than other similar publications and yet do not make any claims that are not thoroughly supported by their data. The methods are clear, if not over-concise, and thus could easily be replicated with a similar data set, or any other data set. The figures are relevant, understandable, and clean. I recommend recommend publication.

Comments, criticisms, and suggestions

line 22 - species diversification might need a qualifier which species, or samples this refers to.

We have shortened and modified the abstract.

line 38 - species concept should be plural

Done.

Somewhere in the Intro and discussion: The fractionation of the scientific community is striking. Key papers that I would have cited in discussing recombination and microbial species are Retchless and Lawrence (<https://www.science.org/doi/abs/10.1126/science.1144876> - they predate the cited Polz paper by a decade), Dykhuisen and Green (<https://pubmed.ncbi.nlm.nih.gov/1938920/>), and Lawrence at al. (<https://pubmed.ncbi.nlm.nih.gov/12446813/>).

We have added the references accordingly (Lines 54, 310, 330).

line 82 – add "in eukaryotes: to read remains unresolved in eukaryotes. Also, I suspect this may be different for allo- and sympatric speciation.

line 87 – Should probably have a citation

line 103 – Probably should be "may consist of" or "according to our methods/analysis may consist of"

Done.

line 109 and elsewhere (e.g., line 728) – most of the data were not available to this reviewer. The accession number in Table S1 do not yet work. Also, the whole genome alignment should be made available in electronic form.

We would like to release the accessions if the paper is accepted or shortly before. We included all the data and files involved in the analyses at GitHub and Figshare. We hope that suffice for now. We added new Supplementary Figures and Tables initially deposited on GitHub. All data stored on GitHub should be available now. The whole genome alignments used in the study have been published at <https://doi.org/10.6084/m9.figshare.24710961.v1>.

line 110 - publically should read publicly

Done.

Figure 1: has no scale bar. Also are enough differences between the highly related strains. Also, this used amino acids, or? This may not be a good choice for closely related OTUs.

The scale bar and information that amino acids were used have been added. We have also used nucleotide sites for phylogenetic constructions, although it was dataset III consisting of only our *Microcoleus* isolates. The topology of the tree did not significantly change, even for the closely related isolates.

line 149 – I am no expert in the Bayesian Binary MCMC (BBM) model, but is seems strange to me that the deepest node has a high probability for area F, whereas a simple parsimony approach would give it about 50% F

The Bayesian Binary MCMC (BBM) method calculates the probabilities of ancestral ranges using the probability of each unit area generated by MrBayes (10.1093/molbev/msz257). It allows for the incorporation of more complex models than simpler parsimony methods. The deepest node has a 94.04% probability for area F (Africa) because all the strains in the deepest branches were isolated from Africa. The BBM analysis also reported that the deepest node has 4.11% for area B (North America) and 1.16% for area A (Europe).

line 153 - It seems to me that there are really two groups of divergence, one for distances below 2000 km, the other for the larger distances. It would be worthwhile to repeat the correlation analysis for only the larger distances.

We repeated the correlation analysis for the groups below 2000 km and found a statistically significant correlation ($r = 0.195$, $p\text{-value} < 0.001$; refer to the Figure below). This finding suggests that a pattern of isolation-by-distance persists even at shorter distances within *Microcoleus*. However, this pattern becomes more pronounced when considering larger distances

as well. Since these results do not significantly affect our observations, we kept the correlation analysis as previously performed.

line 162: "ML tree from SNPs" is this from all the SNPs in the genomes, or just the single copy orthologues?

The ML tree has been reconstructed from all the SNPs in the whole genomes, including non-coding regions. We have changed the text accordingly (Line 144).

line 164 - "likely due to incomplete lineage sorting" – why not gene transfer?

We have added "or gene flow" as well (Line 148).

line 178 – Populations? Species? Groups?

We have modified the text throughout (see the answer to reviewer #2), and lineages have been put here.

line 197: "Statistically positive Tajima's D values were observed in four species (M3, M4, M7, and M10), which suggests that species might be under the balancing selection with signatures of population shrinkage"

These species are marked with (E) on Fig 2a as "expanding", but the Tajima's D suggested the population is shrinking. However, the Bayesian skyline plots suggest the population events in Fig2a are accurate.

In this instance, we have chosen to represent the results of Bayesian Skyline plots (BSP) for the demographic changes instead of Tajima's D or Fu's F . We think adding additional information on

neutrality statistics in Figure 2 would be more confusing. We chose to represent Bayesian skyline plots in Figure 2 because the BSPs are more informative of the demographic history of species (see 10.1534/genetics.108.094904; 10.1093/jhered/esv020) than neutrality statistics. We now specify in Figure 2 legend that the letters denoting demographic events are the ones estimated with Bayesian Skyline plots.

Although both BSP and neutrality statistics could have erroneous inferences about population demography, they give us further valuable information on the dynamics of *Microcoleus* populations. We have modified a paragraph in the discussion where we emphasize that while the BSP method might be a more powerful method of inferring demographic events, there are certain limitations, and the results must be taken with caution (from Line 450).

line 217 - Roary, like most other pan genome assembly pipelines, has the tendency to inflate the pan-genome, especially the cloud and shell genomes. This would be worth to comment on in the discussion. see fig 2 in <https://genomebiology.biomedcentral.com/articles/10.1186/s13059-020-02090-4>

We thank the reviewer for the suggestion. We have added this point to the discussion (Lines 407-409).

Table 1. The designation of location is not clear. The vertical lines in the column appear to suggest 3 groups, but the third one doesn't have a name.

There was an issue when inserting the Table from an Excel format into the Word document, and the Table was split in two. There were only two groups – core and flexible genome. Nevertheless, we have revised the Table according to the reviewers' #4 comments.

Line 229-230: it may be more informative to say of those genes showing association/disassociation what part of the pan-genome they belong to (soft core/shell/cloud)

The program Coinfinder (10.1099/mgen.0.000338) removes genes present in every genome – core genes and those present in very few – cloud genes, as a small number of genes does not yield significant (dis)associations. We have modified the sentence to improve the clarity (Line 209).

line 264 - Reference for phylogenetic signal test is missing.

Done (Line 254).

*line 389: "high ratios of r/m (mean 0.38-2.0; Fig. 3C) between species indicate that gene flow has been important for the introduction of SNPs in *Microcoleus* more than mutation"
If the ratio is below 1 does that mean in some cases mutation is more important for SNP introduction. Also, recombination can occur within a genome, i.e., it doesn't necessarily reflect gene flow/HGT.*

According to Vos & Didelot (10.1038/ismej.2008.93), if the threshold of the r/m value is lower than 1, the species is more clonal than recombinant. Moreover, the value of ρ/θ indicates how often recombination events occur relative to mutation (if $\rho/\theta = 2$, then recombination occurs twice as often as mutations). However, ρ/θ value does not consider either the length of imported fragments or nucleotide diversity (see 10.1038/ismej.2008.93). Hence, these authors proposed using the r/m value to compare the importance of recombination on the diversification of populations, even though these thresholds are arbitrary.

In our case, it is evident that some *Microcoleus* lineages have more than 50% of the genome subjected to recombination. Lower values of ρ/θ indicate that mutation was found to be the predominant evolutionary process in *Microcoleus* lineages. However, three lineages had higher values of r/m , which indicates that recombination contributes to genetic diversity more than mutation. In other words, both recombination and mutation can generate genetic diversity across *Microcoleus* lineages. We expanded our explanation more in the discussion (Lines 353-358).

In the introduction of the paper, we commented: "Building upon Stankowski & Ravinet's definition of the speciation continuum, it can be adapted to microbes and defined as a continuum of barriers to gene flow, where gene flow encompasses the transfer of DNA material realized by homologous recombination (HR)." In other words, by gene flow, we understand here only transfers realized by homologous recombination, regardless of whether they occur through horizontal gene transfer or within a genome. We have now replaced the term "gene flow" with "HR (homologous recombination)" to make it more straightforward (Line 354).

line 402 - Geographical distance be mentioned and discussed here? I might have missed this, but a listing of locations for each "species" would be helpful for this discussion.

We have included a list of each strain's geographic areas of origin (continents/regions) (see Supplementary Data 1). Coordinates of the locations are available as a table in the GitHub repository as they are more important for conducting mantel tests.

We now include geographical distances here, but we will still discuss them later so as not to make the paragraph too long (Line 377).

line 404: "Barriers to gene flow might also arise due to sequence divergence between species, CRISPR repeats," do any of the CRISPR spacers target other Microcoleus species? if so the argument in this paragraph would be stronger, and pinpoint the erection of gene flow barriers.

We are unaware of the research exploring whether CRISPR spacers target other *Microcoleus* species. Nevertheless, we have removed CRISPR analysis from our manuscript. We found it did not significantly contribute to the discussion, and in our previous version, we might have overly interpreted the results.

line 436 "requires fully developed barriers" it is a gradient in recombination frequency. And for

most prokaryotes a pretty steep one. "Fully diverged" is a figment of the imagination for most bacteria, and apparently not even true for most eukaryotes either.

We completely agree with the reviewer. In this context, we emphasize "fully developed barriers to gene flow" to understand the Biological Species Concept's strict stance on genetic isolation without any permeability in these barriers. The terms employed in this paper, such as "fully separate/diverged species", are placed in the discussion to address the scenario where evidence indicates the emergence of various barriers (ecological, genetic, physiologic, etc.) that impede gene flow and relative divergence values (F_{ST}) approaching 1 – full divergence. If a larger portion of the genome exhibits distinctions between lineages and these barriers are evident, then the lineages are on distinct evolutionary trajectories. This would align with the framework of the Biological Species Concept, where higher levels of gene flow within the lineages maintain them in cohesive genetic groups that we can call species.

line 517 – "Ecological differentiation is essential during the early stages of speciation." Only in the presence of gene flow. In case of geographical isolation not.

We have modified the discussion following all reviewers' comments. This sentence has been removed.

Figure 4: The coloring in triangle shaped tree remains unclear. Within each group after a barrier to gene flow is erected, gene flow is high.

Note: We have added a new figure to our manuscript, and the previous Fig. 4 is now Fig. 5.

Figure 5b is supposed to be read from top to bottom. That means there is high gene flow and low F_{ST} at the initial stages of speciation. As we get to later stages, barriers to gene flow arise, and gene flow there is low, while F_{ST} is high.

Collection bias should be mentioned -- many of the older soil samples came from a few locations.

We have added collection bias in the shortcomings section of the discussion (Line 466).

line 630: what were the parameters for prokka annotation? Independent runs of prokka (i.e., not trained on a model) can affect gene calling (especially if the group contains multiple species) and thus inflate the size of the pan-genome.

We have used default parameters for prokka annotation using the bacterial database. The reason for using a bacterial database rather than a specific one for our strains (such as *Oscillatoria nigro-viridis*) is due to the large diversity of our dataset. Moreover, *Microcoleus* is not a model organism.

We have added the information on parameters used for prokka annotation (Line 509).

Reviewer #4 (Remarks to the Author):

The manuscript from Stanojković et al. entitled "The global speciation continuum of the cyanobacterium Microcoleus" is a well written manuscript in which authors sequenced 202 isolated strains and 8 herbarium species classified as cyanobacterial Microcoleus and defined monophyletic discrete groups as "species" in different stages of speciation in a global continuum of speciation processes framework.

The new data and the approximation is well performed, therefore the main conclusions are valid. Furthermore, the significance of this study is mainly related with a notable sequencing effort of the 202 isolates and 8 herbarium genomes, in which the introduction of the continuum speciation process in a widely distributed cyanobacterial genus are the main noteworthy results for further studies in prokaryotes. This work will be of significance in the topic of microbial speciation. The methodology is well explained, however as this study is a follow up from a previous one, some data should be better presented for readers that are new in this model cyanobacterium.

The analytical approach is consistent, however as much of the analyses depend on the "species" definition made by authors, this definition should be stronger, and this reviewer thinks that some complementary data and approaches could be helpful to sustain the further analyses. It is not understood why authors did not use the GTDB-tk framework to compare their results.

The environmental information used to delineate niche preferences is not found in the manuscript and is cited from a previous article from the same group. Here, appears a question: How did the authors selected these 202 strains from the total set of 495 isolated ones from the previous article?.

The selection of 202 strains for genome sequencing followed guidelines for population genomic and general reverse ecology approach (e.g., 10.1016/j.cell.2019.06.033; 10.1101/cshperspect.a018143; 10.1016/j.algal.2023.103128). The first step is to obtain isolates on a global and local scale. We gathered 76 samples of soil and puddles from all continents except South America. Then, from each sample, we isolated 1-11 strains of *Microcoleus vaginatus*, reaching 495 strains altogether. To confirm that strains belong to *Microcoleus vaginatus*, we sequenced 16S rRNA and 16S-23S ITS and found 13 monophyletic clades/lineages that might represent novel species. Thus, this study served as a priori information on the population genetic structure of *Microcoleus* species (10.1080/09670262.2021.2007420).

As there are still no explicit evaluations on minimal sample size for the bacterial populations, we have tried to sequence genomes of as many *Microcoleus* isolates as possible from each of the 13 monophyletic lineages. We considered the availability of the culture and its biomass for genome sequencing as well as choosing isolates to recreate the population's genetic structure from the previous study. Some of the lineages encompassed from 4 to 149 strains; thus, we sequenced whole genomes to have enough representatives of each lineage for population genomic analyses, eventually reaching 202 strains.

We have also included GTDB-tk classification in the revised version of the manuscript (see the comment below).

Furthermore, the determination of the split divergence ages is widely discussed across the manuscript, but the window sizes are too wide to sustain these affirmations. Also, the collection dates of the herbarium specimens could be useful in the BEAST analyses.

We have addressed molecular dating in a previous response to reviewer #1. It is an interesting idea to include the collection dates of herbarium specimens that we have sequenced and we considered it. However, this approach is more suitable for fast mutating viruses, where it is often used to analyze the events happening in months, years, or decades. We are looking at time scales of millions of years. Therefore, the age of herbariums would not provide resolution at this scale.

It is understood that the NCBI data is not still publicly available, and that the accession numbers are available as seen in the supplementary information. However, I would kindly ask you to release the genomes soon.

We would like to release the accessions if the paper is accepted or shortly before. We included all the data and files involved in the analyses at GitHub and Figshare. We hope that suffices for now.

The access URLs to the github repository are not available and in some cases the data is cited to the repository when could be shown as supplementary figures.

We thank the reviewer for bringing this to our attention. The URLs to the GitHub and Figshare repositories should be accessible now. While we thought some figures/tables would be too large to be included in the Supplementary Information, we revised and summarized them. We have added new supplementary figures and tables.

Added Supplementary Data: 3, 7, 8, 9, 10, 14, and 16

Added Supplementary Figures: 8 and 10

Added Figure: 3

While most of the manuscript is clear, other parts need improvements. The introduction is well performed and takes into account several evolutionary theories. The results are well explained, but figures need improvements. The discussion is too long, it should be splitted in ubsections with clear headings and authors should reduce redudancy regarding the introduction and results. References are well suited.

The expertise of this reviewer is mainly regarding genome sequencing, database submission, comparative genomics and phylogenomic analyses, while the usage of different metrics to determine genetic divergence are more superficially understood. Several suggestions to improve analyses and new analyses are also proposed by the reviewer. The english writting is well

understood and the reviewer is not able to further address corrections in that issue.,

The opinion of this reviewer is that this article is worth of publication in the Nature Communications journal, but it needs some major revisions which will be outlined below.

Major comments:

Lines 170-178: This paragraph is essential to justify the further analyses. Is there a review or benchmark analysis which compares these different methodologies?. For this reviewer, the explanation of choosing 13 clusters sound like a parsimonious decision based on the results of different software rather than a justified decision. Is there a bias in the clustering by numbers of genomes (less clusters with singletons)?. While authors presented the values of ANI and established that there are 37 clusters with values over 95 %, is in agreement with the ranges of ANI values reported 86.94-99.9%, and could reflect variable diversification rates (for example comparing the median and standard deviation of intraspecific ANI values for each cluster).

We understand the reviewers' concerns about the designation of isolates to clusters/species, and it is a challenging task with novel isolates and lineages. There are established methodologies and benchmark analyses to identify species present in databases such as GTDB or for the identification of species for metagenomic sequencing. However, we work with species, lineages, and populations not analyzed before (the GTDB analysis also supports this claim; see below). We employed population-level genome sequencing and gathered ecological data for each strain. We had to identify the lineages de novo, which is a challenging task, especially in the context of the speciation continuum.

We followed frameworks commonly used in the non-model organisms. (1) We performed phylogenomic analyses based on several datasets, which were consistent. (2) We defined population structure using several approaches. (3) We applied the modified version of the biological species concept to define the putative species – lineages along the speciation continuum by Stankowski & Ravinet (10.1111/evo.14215). This way, we found the continuum of divergences among the lineages/species as well as a continuum of gene flow.

Considering the singletons. The fastBAPS unoptimized revealed five singletons in comparison to fastBAPS optimized with one, and snapclust revealed none. If we compare fastBAPS optimized and subtract the singletons, we will get only one population difference between the two algorithms. Still, we get to the number of populations of 13 or 14. Thus, the number of populations (excluding singletons) was consistent in all analyses. The only issue remained – where to draw the line between lineages. We applied the monophyletic criterion to resolve this issue.

As suggested, we investigated the diversification rates based on the phylogenetic reconstruction (dataset II). We used the Misse algorithm (10.1111/evo.14517). We found that the net diversification rates indeed largely varied across the whole investigated dataset. We did not include the results in the manuscript due to its length.

Based on this reviewer's comment and the next one, we have modified the paragraph accordingly (Lines 153-169).

This reviewer suggest at this point to further use the GTDB-tk software (Chaumeil et al. 2022, doi: 10.1093/bioinformatics/btac672) to determine in base of this normalized method (RED values) in delineating thresholds for genomic taxonomic classification. Evemn further, if we have a look into the GTDB web page based on the last R214 database version, we can see that the current Microcoleus genus in the GTDB contain 106 genomes divided in 24 different species. It is known, and said by the GTDB authors, that this software is not the reference tool for systematic definitions, however if these different species have more consistency with the geographical and environmental distribution, it would be also an option for the delineation. Summarizing, it is recomended to compare in all clustering methods the consistency of the clusters regarding featrures as geography, environment, genome size, GC content, and any other features to better justify the selection of the final number of "species". Authors, could change the "species" concept and speak about operational monophyletic clusters for example.

We thank the reviewer for the suggestion. We previously considered using the GTDB, but we work with previously unknown diversity, which was unlikely to be included in the database. In any case, we performed the analysis using GTDB-tk software and included the results in Supplementary Data 4. The results suggest that our strains could not be classified into species. A couple of genomes were classified as *Microcoleus asticus* (3), 19 as *Microcoleus* sp., and the rest (179) could be classified only to the genus level.

We have compared the consistency of the clusters regarding:

- (1) While the strains within lineages followed the clustering pattern according to geography, there were many exceptions, which ultimately suggest various levels of gene flow on large geographic scales between strains. Hence, designating genomes to species according to geography was not possible.
- (2) The strains could not be assigned to species using ecological parameters as well. For instance, the clustering of the strains did not follow the habitat of their origin (soil/puddle).
- (3) GC content varied between 45.23-47.43, and the isolates could not be delineated using this feature.
- (4) Genome size is also variable across strains' phylogeny and assigning these monophyletic clades to "species" according to this feature was not possible.
- (5) As previously clarified, utilizing GTBD-tk did not help to assign strains to species.
- (6) We have found that "species" can be delimited based on different morphological traits, such as cell width, cell length, and the pointiness of the apex cell (calyptra). However, including all this information would be out of the scope of this study (see the response to reviewer #2).
- (7) As mentioned in the paper, we also used ANI, 16S rRNA, and different clustering algorithms to assign strains to "species". As only one (optimized Bayesian clustering

analysis) yielded 13 clusters following the monophyly, we have chosen to cluster them more conservatively into this number of groups. However, the ANI would serve as a good tool to identify a new genome and place it into the *Microcoleus* species.

As also proposed by reviewer #2, we now refer to *Microcoleus* species as lineages until the point in the discussion when we offer evidence for calling some lineages species and others not (Lines 327-337).

Table 1: This Table present most of the results regardin selection in genes. However, some field are not well understood. For example, the Localization should be given for any of these genes, understanding that the presence of the gene and the comparison of the multiple sequences helps to determine the neutrality index. Therefore, the Localization (classification) should bve givern for all and also the number of genomes in which is present. The Function column could be more useful if it follows a classification system like that for COG or KEGG classification. Gene IDs could be named as Gene abbreviation. Gene ID could be easier to identify for a potential reader, if the NCBI protein accesion ID could be given for a representative sequence of the cluster. It is not know by the reviewer if the Table can have colors in the Journal forming.

We have revised the Table 1 and now we include gene ID (from the NCBI), gene abbreviation and gene name (from prokka annotation), localization including the number of genomes in which genes are present, and COG classification according to the eggNOG-mapper (16 genes were missing KEGG annotation). We have removed the colors.

Discussion: The discussion is too long and repetitive, it was hard to rerad for this reviewer and some (not mandatory) minor suggestion were done regarding each paragraph. It is not known if authors can join results and discussion in the results section and leave the two last paragraphs which explains the model and the drawbacks as a general discussion.

We have modified and shortened the discussion following suggestions given by all reviewers.

Highlighted are the parts of the discussion that have been changed, while the rest of the discussion remains unhighlighted despite reordering paragraphs. We think the changes are more visible this way because otherwise, we would have the whole discussion highlighted.

Figure 1. The phylogenomic tree should have more preponderance in the figure 1. The phylogenomic tree reconstruction need to show the support of the different nodes and a scale of the distances. The coloring of the different continents is not so neccesary as they are distributed in different columns (this is only a personal comment, as there is redundant information). The color strips occupy too mucch space. This figure could be used also to show which genomes are new in this study and also information of the environments. Authors can easily manage and add this information in the Itol webserver. In panel B, authors should briefbly explain the Y-axis or how the genetic distance was determined between each pair of genomes and also explain which dataset was used.

We agree with the reviewer that the coloring of different continents was redundant information; hence, we revised Figure 1 accordingly. We now show the scale of the distances with support of different nodes (nodes with red points denoting bootstrap support ≥ 99). Geographic areas are presented in different columns, and the color strips now correspond to the habitats from which strains have been isolated. Additionally, in black boxes are genomes added from GenBank. Panel b now includes information on genetic distance. We left in the legend information on the dataset used for this figure and analysis.

The differences in the genome size between the different isolates is noteworthy, however it is not addressed in the manuscript because most of these differences are given mainly in the flexible genome which is a huge part of the genus pangenome.

Here also appears something that could have (or not) importance in the differences in the pangenomes and genome sizes which are the mobile genetic elements. As these elements are thought to enhance the adaptability to different or harsh environments, maybe watching the differences could be important for the different species. Authors could use common tools to determine over the assemblies the presence of these elements: geNomad (<https://github.com/apcamargo/genomad>), MOBsuite (<https://github.com/phac-nml/mob-suite>), or Platon (<https://github.com/oschwengers/platon>).

The genome size differences have not been discussed in the previous version of the manuscript; however, we are now discussing the genome size variability in the context of HGT, as proposed in the following comment (see answer below this one).

We thank the reviewer for the suggestions. We utilized the proposed tool Platon to investigate the presence of mobile genetic elements (MGEs) across all *Microcoleus* strains, and we have observed their consistent occurrence (data available). While the number of MGEs per strain would be informative, we acknowledge that a more in-depth understanding could be gained through functional annotation. This step would allow us to discern the potential functions of these elements and their significance to the adaptation of *Microcoleus*. While it would be interesting to proceed with the functional annotation, we believe that integrating this analysis would only contribute to even more supplementary materials, which are already extensive and may not enhance our current discussion substantially.

In light of this, it would be more fitting to include the proposed analysis in our upcoming research when we have third-generation sequencing data and focus on transcriptomics and physiological responses of *Microcoleus* strains under various environmental conditions (such as desiccation).

It was also suggested somewhere by this reviewer, that the effect of HGT or terminal paralogous (gene duplication) in the speciation process could be determined to see other influencing factors and to balance the role of them against recombination and mutation. In the Orthofinder results can be obtained the paralogous sequences, and there are several tools like HGTector

(<https://github.com/qiyunlab/HGTector>) to determine genes potentially affected by this process.

We appreciate the reviewer's suggestion to perform additional analysis on HGT. We have utilized the proposed tool HGTector and found that many genes underwent HGT (150,883). Of these genes, the majority have a cyanobacterial origin, and the number of HGTs significantly differed between different *Microcoleus* lineages. Moreover, the number of HGTs differed between strains originating from different habitats (puddle – soil). Lastly, we have conducted a correlation analysis between the genome size and the number of HGTs detected in each lineage and found a significant correlation (see Figure 3; Supplementary Data 8, 9, and 10). We have included these analyses and a new Figure 3 in the revised version of this manuscript.

Similarly to our previous response for MGEs, we think the functional annotation of the genes affected by the HGT would be informative here. However, as explained in the previous comment, this analysis would not significantly enhance our current discussion and would be more fitting in with our upcoming research.

Minor comments:

Abstract

line 13: This phrase is unclear about what is driven.

We have now shortened our abstract and rephrased some parts.

**lines 16-17: This phrase is strange in the context of the study, specifically speaking of the *M. vaginatus* species, because in Table S2 there is only one *Microcoleus vaginatus* genome, while the others are unknown species.*

According to phylogenetic analyses, all the genomes belong to the *Microcoleus vaginatus* clade. Many of the submitted genomes to the GenBank can be found only by *Microcoleus* sp., when they are actually one of the species within the *Microcoleus vaginatus* complex. The genus requires further taxonomic identification and proper characterization of species. We are currently working on it.

line 25: Biosynthetic processes is not a correct term, as much part of the metabolism is biosynthetic. Do you refer to secondary metabolism biosynthesis?

Yes, we did refer to the secondary metabolism. We have modified the sentence accordingly (Line 22).

Introduction

lines 52-54: This phrase is hard to digest, as in the previous phrase authors speak about the continuum of barriers (similar to accumulation of barriers) and in this phrase about the strength.

Furthermore, the cryptic species is a not well defined concept, therefore mentioning here open question which will not be adressed.

lines 54-55: This phrase doesn't have relevance in the context of genetic differences, as probably you refer to the phenotype as the morphology considering the two cited references; while the microbial world is featured by extensive genetic diversity with similar morphologies.

We have removed the sentence as it was redundant, as well as the following one.

lines 55-57: Please provide references, or this phrase is supported by those on the next phrase?.

This phrase is supported by studies listed in the following sentence.

lines 57 - 59: Isolated in which terms, geographically or genetically?

We have added "genetically and ecologically" (Line 49).

Line 61: It is understood that authors don't mention in the manuscript the Horizontal Gene Transfer mechanism as having roles in diversification because when comparing at genus level, the influence of this mechanisms could be smaller than others, however it would be worth mentioning it (and if it's possible to perform some analyses to compare with the influence of recombination).

We have now added horizontal gene transfer as well.

line 77: Please fix the parenthesis.

Done.

lines 87-88: It is known the expansion of freshwater (origins of Gloeobacter) and then marine cyanobacteria regarding the GOE, however it is not known for this reviewer if there is evidence of the importance of cyanobacteria in terrestrial systems in those time scales.

We modified the sentence to be more conservative. Interestingly, we looked at this question before in our paper (10.1111/mec.12948). We found several early diverging clades composed of terrestrial cyanobacteria before, during, and shortly after GOE. Thus, we could suspect that terrestrial cyanobacteria were significant during the mentioned time.

Results

lines 121-125: While in Table SI is shown the high degree of completeness and low

contamination of the genomes (lower for the herbarium specimens), it would be better to report here the mean and minimum values of those metrics. Also, it is noted the huge difference of the genome size, the lower genome size obtained is for a lower complete genome?.

We are now including the mean and minimum completeness and contamination scores (Lines 106-109). There was no significant correlation between the genome size and the completeness scores across all strains ($r = 0.078$, $p = 0.25$).

lines 125-126: The features of the genomes from the NCBI also should be determined, and only genomes with high completeness and quality should be retained for further analyses.

We have also performed analyses for genomes from the NCBI using CheckM and presented them in Supplementary Data 1. We have retained all the genomes as previously used, as they do not significantly affect our observations.

line 128-129: Please provide somewhere which 30 genomes were used in dataset I.

We have put an asterisk after the strain name in Supplementary Data 1 to mark genomes used in Dataset I.

line 136-137: Please explain somewhere why this strain was used as outgroup.

We sequenced the strain that appears to be most closely related to the *Microcoleus vaginatus* clade, and it consistently got placed on species trees into a separate clade/branch (Supplementary Fig. 1, 3, and 5). The next closest strain is *Kamptonema (Oscillatoria)* PCC 6506, but the gap between *Kamptonema* and *Microcoleus* clade is too deep. We have added an explanation in Line 122.

lines 144-146: Please provide the accession numbers for the Genbank genomes in the Figure S1. For example, why are two genomes as the PCC 6506?. This is also related with the fact that in Table S2, the names of all genomes of the dataset II were changed, however it should be better to show the original names.

Accession numbers of all genomes used for Supplementary Fig. 1 are provided in Supplementary Data 2. We are also now showing the original names found in the GenBank.

Supplementary Fig. 1 indeed included two genomes as PCC 6506, so we removed one (*Kamptonema*) and made a new figure.

lines 144-146: It is not understood which parameter was used to define the M2_D5 genome as sister taxa and not as part of the *Microcoleus* continuum. The results for the ancestral geographical results could also change if this strain is included inside the *Microcoleus*, being a potential African origin as outlined in the text. Please justify.

We have chosen the outgroup M2_D5 as it was consistently placed in the phylogeny as the most closely related to the *Microcoleus vaginatus* clade (Supplementary Fig. 1). It also might belong to the continuum of *Microcoleus* species, but we decided to keep it outside of the analysis (together with the strain N3_A4 belonging to the M13 lineage) because we lack population-level data for further analyses. Collecting more samples and isolating and sequencing more genomes of *Microcoleus* would allow one to investigate whether M2_D5 is a part of the continuum of *Microcoleus* species or not. However, the strain M2_D5 was used for the phylogenomic analyses (Supplementary Figs. 1, 3, and 5), including ancestral state reconstruction of the geographic origin (Supplementary Fig. 2).

lines 166-168: Please, explain better this sampling grouping for readers not aware of the methodologies used in the previous study.

In order to clarify the grouping, we have added a new supplementary table (see Supplementary Data 3), where we have the sample names and the information on how many lineages were found from each.

line 217: Authors used Orthofinder and Roary software over the same dataset, are the results consistent between both?

We did not perform pangenome analyses using Orthofinder because this pipeline is designed for phylogeny. In any case, we looked at the topology based on the core genome alignment using both Orthofinder and Roary. The ML tree topology produced based on both datasets was consistent.

lines 221-223: do you mean clusters of genes (be careful when you say CDS)? The number of clusters of genes being in the core/soft-core is too close with the early determinations on cyanobacteria (Simm et al. 2015, doi: 10.3389/fmicb.2015.00219). Is there an effect in these numbers of the low-completeness genomes?.

We have removed CDS and modified sentences accordingly. All the genomes (201) used for pangenome reconstruction had an average completeness of 99.5%. There was one genome with a completeness score of 90.99% (K4_C2), one with 95.74% (Z1_A1), seven with completeness between 97-98.9%, while the rest had scores of completeness higher than 99%. Having the majority of the genomes with high completeness scores, we believe that the impact of the slightly lower completeness might have only a limited impact on the roary detection of core/softcore genes.

lines 228-230: what do you mean with metabolic genes and with the "annotated" in parenthesis?. The URL is not available. Authors also could show these results as supplementary tables.

We have now kept only the number of genes found to be showing significant association/disassociation (Line 210). We have also included the list of these genes in a new Supplementary Data 7.

lines 231-233: please be more specific in which genes are you referring to. One of the functions is biosynthesis, please be more specific.

We have now specified that it is the "biosynthesis of secondary metabolites" (Line 213).

lines 242-246: maybe is in discussion, however are you referring to a potential absence here of speciation islands as defined in introduction? Please authors can make an effort in summarizing the data in supplementary tables or figures.

We have added a point about the lack of genomic islands in the discussion (from Line 376). Additionally, we summarized all the data and added new supplementary figures and tables (listed in our response at the beginning).

lines 251-252: Besides the tests, please provide the raw environmental parameters. Which are all the variables measured in the original samples?

Raw environmental parameters have been included in the GitHub repository (URL available now) to avoid further increase of supplementary tables. Moreover, our previous study has already published the raw environmental parameters for each strain. None of the variables has been measured from the original sampling sites. All parameters have been acquired from publicly available datasets based on the geographic location of the strains.

lines 255-259: These reported tests are those only for variables with statistical differences or authors did find differences in the other environmental parameters?

We selected the environmental variables with the strongest correlations from the mantel analysis, including (1) temperature, (2) precipitation, (3) solar radiation, and (4) soil characteristics. Specifically, we identified the strongest correlations from a pool of multiple variables under each environmental factor (e.g., six parameters related to temperature, four to precipitation, 11 to soil characteristics). We then represented variables with the strongest correlations in Fig. 2 (see Supplementary Data 13 for all variables).

line 272: Microcoleus species.

Modified to "Microcoleus lineages" (Line 262).

***lines 272-279: *These results are good, maybe they should have further protagonism?.*

We appreciate acclaim very much. However, we have decided to remain conservative with interpreting the results from the phylogenetic signal and Mantel tests as was proposed by reviewer #1. We would need to perform further analyses on more environmental variables with physiological experiments to confirm the role of certain variables in the genetic diversity of *Microcoleus* species.

lines 289-290: *"estimates of isolated in each species..." this phrase sounds strange.*

Modified to "isolates" (Line 280).

lines 290-292: *This phrase should be fixed as the Kruskal-Wallis test determines if there is any difference between the groups testes, and in this phrase it reads that every group is different from any other.*

We have now changed "between any two groups" to "between some lineages" (Line 282).

lines 305-306: *Is there any bias given by the level of assembly of the CRISPR regions? This question is regarding the difficulties on assembling these genomic regions. The CRISPR repeats are the same for all the genomes in this dataset?.*

Bias may exist due to the assembly level of CRISPR regions. However, as previously addressed in response to reviewer #3, we have decided to remove the CRISPR analysis. There might have been an excessive interpretation of the results in the previous manuscript version (discussion related to CRISPR), so we decided to remain more conservative.

lines 313-314: *This URL is not available, and it seems related with the Gubbins results rather than with the CRISPR analyses. Here a supplementary table with the numbers of CRISPR repeats and the values of recombination blocks would be helpful for readers. As suggested above, here authors could show the CRISPR sequences, and if the repetition of the different arrays is different across species.*

These analyses have been removed from this manuscript version (see previous comment).

lines 320-321: *Authors say that in four regions are concentrated the mentioned genes; however, besides recombination, it is not mentioned or analyzed if the *Microcoleus* genomes have extensive genome rearrangements (which could reduce the genomic regions common in the dataset) or if they maintain as much as possible the genome synteny.*

Indeed, we expect the genome rearrangements to be as common as in other bacteria. However, this paper mainly focused on the population genomic diversity. We plan to focus on genome rearrangements and structural variation with the long-read genome assemblies, which will help us shed some light on the evolution of the genome rearrangements in the *Microcoleus*.

lines 324-326: please read the commentary about the Table 1 regarding the function assignment. This is related with the topic that biosynthesis is a wide metabolic function, and maybe authors want to focus on secondary metabolite biosynthesis.

We have addressed this in one of the earlier comments in Table 1.

lines 326-330: The figure S9 (S11 doesn't exist) need more detail about which genome was used to map de NI values. This figure shows, as well the text is explicit, that the >1 NI values are across all the genomes, therefore it is not understood how the previously mentioned 4 regions were determined and in which genome. It is recommended to add a supplementary Table instead a hyperlink to the github repository.

We have corrected the numbering of supplementary figures. We understand the reviewer's point about not being clear about how NI values were mapped. We have previously shown all the NI values for all the genes (statistically significant based on the MK test) mapped over the reference genome of *Oscillatoria nigro-viridis* PCC 7112. We have modified this supplementary figure and show each lineage separately, with NI values of significant genes mapped over the reference genome (Supplementary Fig. 10a-1).

In our response to reviewer #1, we explained that detected windows of elevated differentiation were found between the following pairwise lineage comparisons (between M2 and M8, M12; between M4 and M6, M9, M10, M11, M12; between M5 and M10; between M6 and M7, M9, M10). In other words, one region of elevated differentiation does not appear across all the lineages. In Supplementary Data 11, we highlighted only genomic regions with elevated values of both F_{ST} and D_{XY} . In the Methods (Lines 654-656), we explain how only genomic regions selected had the top 1% of both F_{ST} and D_{XY} values in pairwise lineage comparison.

Discussion

lines 336-348: This introductory paragraph to the discussion, should be shortened and as it tries to summarize the findings of the study and many of this information will be taken into account later.

We have removed this paragraph from the discussion.

lines 338-339: "a continuum of *Microcoleus* species..." the continuum is referred across the manuscript as regarding the speciation process, please clarify.

The continuum has been mentioned in the context of the speciation process. However, we mentioned in the introduction that the continuum of *Microcoleus* species actually represents *Microcoleus* species at varying stages of divergence along the speciation continuum (Lines 84-86) due to genetic and ecological differences between them. This paragraph was removed, as previously mentioned.

lines 360-374: this paragraph should be joined in a section specific to the species and continuum speciation process. Authors should shorten or refocus the information of this paragraph that is given in sections above (lines 365-368)

A big part of this paragraph has been removed, and one sentence (see below) has been modified and moved to the following paragraph – Line 327.

The sentence moved and modified: "We captured all the features necessary to characterize some of the *Microcoleus* lineages as species and place them along the continuum, including preferential gene flow within the groups and variations in the extent of geographic, ecological, and genetic barriers impeding gene flow^{8,57}."

lines 375-385: the main message of this paragraph is that of lines 379-381, authors can shorten the other phrases (and provide a supplementary table with the Fst analyses).

We have modified this paragraph (see Lines 338-352) and added HGT results as well as the explanation for the lack of "islands" as previously suggested (Lines 344-352).

lines 387-340: the main message of this paragraph is between lines 387-394, it is suggested to shorten the other phrases.

We have shortened and modified this paragraph following the suggestion of reviewer #3 as well (Lines 353-359).

lines 402-420: this paragraph is too much long and it mixes too much information. It is not understood the role of natural competence in the rates of homologous recombination, and the CRISPR-cas seem not to be barriers to this process; therefore the topics seem to be intermixed but not justified. Maybe, the discussion of the barriers of recombination due to the loss of the specific 20 bp sequences that are recognized by recombinases could be more informative.

We have modified and removed a part about the barriers from this paragraph. We have also removed the discussion on CRISPRs. We kept the part on homologous recombination and moved it to the following paragraph - Lines 359-368)

lines 423-429: this paragraph could be shortened and joined in a special section regarding the environmental constrains in speciation.

We have joined this paragraph as suggested and modified it accordingly as we added information on HGT as well (see Lines 396-402).

lines 431-445: this paragraph should be joined in the section about the speciation continuum which has been addressed in the beginning of the discussion.

We have moved this paragraph to the beginning of the discussion and modified it accordingly (Lines 308-320). One sentence has been moved to the results (see Lines 275-277).

lines 447-459: this paragraph seem to be more related with results, complementing those analyses rather than properly discussion.

This paragraph has been moved to the beginning of the discussion (see Lines 321-337). We think this paragraph is important to stay in the discussion since here we offer insight into what lineages may represent species and what cannot be considered species yet. We moved one sentence to the results as previously mentioned.

lines 461-499: these three paragraphs could be joined in a section in which a table is presented with the split time divergencies of the different "species" and the climatic trends along with their references. Therefore authors could focus in the main discussion as that of lines 485-487.

We have kept these paragraphs in the discussion for now, modified them accordingly and moved them towards the end (Lines 431-463).

lines 521-526: the first of these two paragraphs is too descriptive, therefore the main message is lost.

We have shortened the paragraph, modified it accordingly and moved it earlier in the discussion (Lines 405-431).

lines 528-575: these paragraphs could be shortened and focused in the ecological drivers of barriers to HR as well specific genes for adaptations to the different niches in which authors detected preferences between different "species".

We have shortened some of these paragraphs and modified them accordingly (from Lines 393-430).

lines 577-586: please provide some references in other models for these different suggestions.

Done (Lines 468 and 474).

Materials and methods

lines 603-604: what about the axenicity or uncyanobacterial state of the cultures? could authors provide the checkM parameters obtained of the assemblies before the binning procedure with maxbin?.

Cyanobacterial cultures are extremely hard to be axenic. We have a unialgal culture collection, meaning a single cyanobacterial species with bacteria is present. MaxBin was used to segregate the genomes belonging to other bacteria and one bin belonging to cyanobacteria.

lines 608-609: please provide the parameters of quality filtering. Also, there is a word missing before "... quality reads". Did authors used the SPAdes software in default mode, or used the --meta or --careful options?.

We used SPAdes in default mode with the option `-isolate`. We previously tested several assemblers for our dataset, including SPAdes with several settings. SPAdes performed the best, and we got the most complete and the least fragmented assemblies using the mentioned settings. We added the parameter used in the methods section (Line 493).

lines 610-612: did authors searched for the rRNAs present in the genomes (and unbinned contigs). Sometimes, even from cultured strains these regions are not binned and valuable information is lost when submitting to NCBI.

We did, using barnap (<https://github.com/tseemann/barnap>), which is incorporated in the prokka on the binned contigs.

line 625: authors are encouraged to release the data before the next round of revisions (if applicable)

All data should be available now.

lines 628-629: while prokka is the gold standar tool to annotate genomes, to further classified the annotated sequences it is rtecccomended to use the eggnog-mapper (<https://github.com/eggnoget/eggnoget>) or deepnog tools (<https://github.com/univieCUBE/deepnog>) to get the COG hierarchichal classification of the genes or the Kofam-KOALA web server (<https://www.genome.jp/tools/kofamkoala/>) to annotate under the KEGG classification.

We employed prokka precisely for the reason mentioned. In any case, we included the COG classification of the genes present in the highly diverged regions in Table 1, where we discuss the gene functions.

line 630: it is not understood by the reviewer why authors used that genome and not the *Microcoleus vaginatus* genomes (GCA_022701275.1 or GCA_000214075.2).

Oscillatoria nigro-viridis PCC 7112 actually represents *Microcoleus vaginatus*, but it has the wrong name in the GenBank and in the Pasteur Culture Collection. We performed a phylogenetic analysis to confirm this. The strain can be found in the Pasteur Culture Collection of Cyanobacteria and represents a reference genome. Cyanobacterial strains often have incorrect species names. We are preparing the taxonomic revisions of the genus, which should help to ameliorate this issue.

Furthermore, GCA_022701275 is contaminated (checked in the GenBank), and GCA_000214075 is incomplete. Both genomes cluster with *Oscillatoria nigro-viridis* PCC 7112 in the same group. *Oscillatoria nigro-viridis* PCC 7112 represents the complete assembly to the chromosome and plasmid level.

lines 652-654: authors can join this with the previous paragraph only outlining the differences in the procedure (or must to cite correctly as in the previous paragraph the software).

The citations have been added accordingly.

line 673: please provide the reference (doi: 10.1093/nar/gkz361).

Done.

line 710: which type of distance measurement was determined with the R package (e. g. geodesic, grat-circle, etc)?

The function has been specified in the text – distGEO, which uses distance measurement geodesic (Line 599).

lines 745-748: how authors determined the CRISPR repeats (remove the word spacer)?

A number of CRISPRs per strain and lineage has been obtained from prokka annotations. Prokka uses a tool to automatically detect CRISPR repeats called CRISPR Recognition Tool (CRT). However, as mentioned previously, we have now removed the CRISPR analyses.

lines 776-790: these methods were not completely understood by reviewer in terms of inputs and outputs. Specially, what was done with the CDS in lines 779-781. What do you refer with "samples" in line 781. In lines 784-787 there is not space given for genes with neutral selection, please explain.

We have corrected the "samples" with "genes", and corrected the wording to read that we used the annotation of *Oscillatoria nigro-viridis* PCC 7112 as the reference genome for the analysis. The space was not given to the genes with neutral selection (i.e., neutrality index $NI = 0$) because this analysis mainly aimed to infer the potential adaptive value of the genes within the region of elevated divergence.

We have used the NI values calculated for all the genes between lineages to map them over the reference genome and show the position of selection genome-wide (now provided as updated Supplementary Figure 10).

Figure S2. Authors should improve the figure legend regarding what do you refer with the bold and red highlightning. Do you mean genomes/strains grouped in different species coming from the same environmental sample?.

We have added a supplementary table (Supplementary Data 3), where we give the samples and the number of *Microcoleus* lineages found in each. This table should help in understanding Supplementary Fig. 2 as well as the text in the results section.

Figure 2: Align the panel letter B) with the others.

Done.

Figure 3: In panel A) and in supplementary table 9 the given values are the mean inside and between species groups?. The standard deviation should be given in the Table (and I suppose that these deviation values are given in figure 3 B).

Note: We have added additional Figure and this is Figure 4 now.

Correct. The standard deviation values are in Figure 4b. We have added another supplementary table with all the values per strain (Supplementary Data 14).

Reviewer #1 (Remarks to the Author):

The authors have addressed some of my comments but I still think that the results are not fully supporting the conclusions of the authors. The results could have alternative explanations and using the molecular clock on the 16S gene is unlikely to give reliable results. I think that the authors should be a bit more cautious with their conclusions.

Reviewer #2 (Remarks to the Author):

The authors adequately addressed my concerns.

Reviewer #3 (Remarks to the Author):

Line 120: "the final dataset III..." the difference between dataset 2 and 3 is unclear as both are stated to be used for phylogenomics. Maybe state explicitly what analysis were done with which.

Line 139: "dispersal of *Microleus* strains was geographically restricted" - I remain unconvinced by the performed analyses, and by the analyses described in the rebuttal. This certainly should be toned down as the correlation value is not that high. Also, the correlations appear to be due to short geographical distances. Should the finding of similar strains found close by be described as a restriction of dispersal? Dispersal takes time, and the local strains might get to more distant locations in the future. Or in other words, describing the observation of a geographical signal in the plot against sequence divergence as restricted might be a poor choice of words.

Line 187 ff. This doesn't look like a "toned-down" discussion of the Skyline plots.

Line 141: "*Microleus vaginatus*" This is the first mention of the species name, previously it was referred to as the *Microleus* genus. Was it determined that the strains are members of the same species before this section? Probably not, the ANI values in Fig 1b have minimum values in the low 80s, so they wouldn't be considered the same species.

Line 150: what does "sample" mean? (as in groups on the tree?)

Line 160: "The pairwise average nucleotide identity (ANI) showed that *Microcoleus* isolates shared 86.94-99.9%". However, Fig 1b (y-axis; 1-ANI values) has many values below 86% (i.e., .18-.20).

Line 210: "In particular, 1,079 genes showed significant associations, while 1,258 were significantly disassociating, .." how was significance determined?

Line 341: "We found no apparent peaks of elevated differentiation (neither 'islands' nor 'continents'; sensu Shapiro & Polz18) across the genome but rather a broad elevation of its diversity (Supplementary Fig. 8) as previously"
Could this not be the result of them being different species instead of being different points in a continuum of speciation?

Line 348: "Given the heterogeneity of soil systems and their selective pressures on organisms, having small genomic regions ('islands') with few specific genes may be evolutionary disadvantageous." I do not follow this logic.

Line 511: "SNPs were detected by freebayes v1.3.299 with..."
Given that the core genome is only 5% of the pan-genome or ~600 genes were SNPs from the flexible genomes used for the phylogenies?

Line 540: " The best-fitting model selected by ModelFinder104 was LG+I+G and branch supports were computed using ultrafast bootstrapping with 2000..."
I'm not asking them to redo all of this but for this dataset specifically (which is the cyanobacterial

phylogeny) site homogeneous models such as LG+ would not be as robust as a CAT or C60 (PMSF) model (site heterogeneous) especially when its 2000 single-copy orthologs and cyanobacterial are quite well known for their complicated phylogenies. Datasets 2 and 3 would probably be fine with LG+ models since they are all the same genus.

Line 578: The 16S phylogeny was not used for dating. How resolved was it? This might be a good thing to include in the supplementary data.

Line 641: Is the assumption of a strict clock justified? Fig. S1 doesn't not look like it. I did not find a link to 16S *Microcoleus* dataset used for the analyses in the manuscript, nor on the linked github and figshare sites.

The text says that the analysis used GTR+G+I. Is GTR appropriate in this case? I assume the tree was calculated in iqtree, which by default provides a compositional analysis, and flags sequences with a significantly different composition.

Line 540: "The best-fitting model selected by ModelFinder104 was LG+I+G and branch supports were computed using ultrafast bootstrapping with 2000..." I'm not asking them to redo all of this but for this dataset specifically (which is the cyanobacterial phylogeny) site homogeneous models such as LG+ would not be as robust as a CAT or C60 (PMSF) model (site heterogeneous) especially when its 2000 single-copy orthologs and cyanobacteria are quite well known for their complicated phylogenies. Datasets 2 and 3 would probably be fine with LG+ models since they are all the same genus.

Reviewer #4 (Remarks to the Author):

The new version of the manuscript from Stanojković and colleagues "The global speciation continuum of the cyanobacterium *Microcoleus*" has improved too much considering that authors took into account most of the main concerns of the reviewers. It is acknowledged that some claims are now more conservative and that the highly controversial topic of "species" was solved with the usage of the "lineages" word. The writing was benefited of shortening some sections and figures/figure legends were improved. The discussion section would benefit of a division in subsections with clear headings. Taking all of this into account, including the previous comments of this reviewer, this manuscript is suggested to be accepted to be published in the Nature Communications journal. However, there are some minor comments which should be addressed but with no necessity of another round of revision according to this reviewer.

Minor comments:

line 93 and elsewhere: please, be consistent with the number of sequenced strains. In some places appears 201 and in others 202.

lines 22 - 225: These differences could be product of the genome size differences. This, as being correlated by genome size in the previous phrases. Authors, could make the direct link watching the differences of number of HGT normalized by each 1000 genes or genome size and seeing if the differences disappear between these lineages. The same for the analyses in the figure 3d (also if the differences are retained, they could be more linked to environmental factors rather than to genome size).

line 243: Supplementary Data 12?

line 281: Please, fix the figure numbers.

line 297: Please fix Supplementary data number.

line 314: Are you referring to Figs. 5a and 5b?

line 337: "different species" maybe sounds better.

line 357: "Role" should be in plural.

line 373: Here are you referin just to the different species mentioned by the UPCEL analysis, or to the 12 lineages?.

line 421: RcsC if authors are referring to the protein.

line 492: Maybe are you referring to "high quality reads".

Fig 1. I don't know if it is a problem of the pdf, but the squares on the habitats legend are not aligned.

Reviewer #1 (Remarks to the Author):

The authors have addressed some of my comments but I still think that the results are not fully supporting the conclusions of the authors. The results could have alternative explanations and using the molecular clock on the 16S gene is unlikely to give reliable results. I think that the authors should be a bit more cautious with their conclusions.

We thank reviewer #1 for their suggestions and have implemented several changes in response to their concerns. Specifically, we have removed the Bayesian Skyline plot analyses, leading to the removal of parts related to demographic changes throughout time, especially those related to the molecular clock. We have also added a more conservative conclusion to the dating analysis and acknowledge its limitations (Lines 428-429 and Lines 451-454).

We agree with the reviewer that analysis using the molecular clock on the 16S rRNA may be unreliable. However, there is no universal molecular clock for prokaryotes, 16S rRNA, or orthologues. A general problem for molecular dating of microbes lies in the absence of clear calibration points and the use of points that have occurred in the distant past (e.g., the emergence of oxygenic photosynthesis; 10.1186/1471-2148-1-4). Regarding cyanobacteria, a reliable estimation of divergence times requires a careful interpretation of the fossil records (that are quite scarce), using multiple fossil records to calibrate points, as well as considering many parameters for selecting a proper evolutionary model. Moreover, specific morphological characters (autapomorphies) are lacking in fossil records (see 10.1016/j.earscirev.2017.10.001), so it is not possible to adequately connect them to extant clades, such as *Microcoleus*.

Although some fossil records are available, they belong to the pleurocapsalean or nostocalean taxa, which display complex branching or specialized structures such as akinetes. Using these characters for calibration is unlikely to provide higher certainty for oscillatorian taxa as their evolutionary rates can greatly differ from akinetes/trichomes of other cyanobacterial groups, and the use of such fossils may introduce even more uncertainties regarding specific, more recent divergence events. Overall, microbial fossils that can be clearly assigned to known species are rare.

As we have explained in the rebuttal, the 16S rRNA is commonly used for dating in prokaryotes due to its known mutation rate, slow rate of sequence divergence, and widespread use in cyanobacterial taxonomy. Moreover, the mutation rate that we applied was specifically developed for cyanobacteria. The core genome would be a possible alternative; however, there is a lack of information on the evolutionary rates of all genes found there. A high variation in divergence rates across the genes and lineages would mean that no single molecule can serve as a universal clock in (cyano)bacteria. Therefore, there is currently no better approach to reliably date cyanobacterial phylogeny. We have now added a point to the methods section, where we explain why we chose 16S rRNA for dating analyses (Lines 555-557) with added limitations to using the gene in dating analyses (Lines 451-454).

Reviewer #2 (Remarks to the Author):

The authors thank reviewer #2 for their positive and concise assessment.

Reviewer #3 (Remarks to the Author):

The authors thank reviewer #3 for their helpful suggestions, and we have implemented changes in response to their concerns.

Line 120: "the final dataset III..." the difference between dataset 2 and 3 is unclear as both are stated to be used for phylogenomics. Maybe state explicitly what analysis were done with which.

Dataset II was used to infer evolutionary relationships between all genomes (sequenced in this study and obtained from GenBank). This phylogeny has then been used to perform ancestral area reconstruction and phylogenetic signal tests. On the other hand, dataset III has been used to infer the species trees (only genomes sequenced in this study for population genomics analyses). Now, we clarified the difference between the two datasets by stating that dataset III was used for the inference of species trees (Line 121).

*Line 139: "dispersal of *Microleus* strains was geographically restricted" - I remain unconvinced by the performed analyses, and by the analyses described in the rebuttal.*

This certainly should be toned down as the correlation value is not that high. Also, the correlations appear to be due to short geographical distances. Should the finding of similar strains found close by be described as a restriction of dispersal? Dispersal takes time, and the local strains might get to more distant locations in the future. Or in other words, describing the observation of a geographical signal in the plot against sequence divergence as restricted might be a poor choice of words.

While we think our results can be interpreted in a way that there is a signal of isolation by distance among *Microcoleus* lineages, we agree with the reviewer that we should be more conservative with our interpretations. Therefore, we have removed the expression 'geographically restricted' and the discussion on lineage dispersal. Instead, we now clarify that the observed correlation suggests geography as a contributory factor to the diversification of *Microcoleus* lineages in the results (Lines 140-141) and discussion sections (Lines 381-382). We highlight that in the discussion section (Lines 447-451), we had already recognized the observed correlation between geography and sequence divergence as a limitation of the study.

Line 187 ff. This doesn't look like a "toned-down" discussion of the Skyline plots.

Due to the concerns expressed by reviewers #1 and #3 in the first and second reviews, we have now removed Bayesian Skyline Plots from the manuscript. We have replaced Supplementary Figure 6 with a new one with Tajima's D and Fu's F boxplots. We have also removed the demographic information initially included in Fig. 2a.

Line 141: "*Microleus vaginatus*" This is the first mention of the species name, previously it was referred to as the *Microleus* genus. Was it determined that the strains are members of the same species before this section? Probably not, the ANI values in Fig 1b have minimum values in the low 80s, so they wouldn't be considered the same species.

By mentioning the species *Microcoleus vaginatus* in this section, we intended to emphasize that our strains had previously been identified as *Microcoleus vaginatus* based on 16S rRNA, 16S-23S ITS, as well as morphology. Also, we wanted to clarify that our isolates actually constitute a much larger complex, representing a global continuum of at least 13 lineages based on all the results outlined in this part (phylogenomics and different clustering analyses). Nevertheless, we agree that mentioning the species name in this segment might cause confusion; thus, we have removed it and kept only the term *Microcoleus* (Line 142).

Line 150: what does "sample" mean? (as in groups on the tree?)

We were referring to the environmental sample. We have modified the sentence accordingly (Line 151).

Line 160: "The pairwise average nucleotide identity (ANI) showed that *Microcoleus* isolates shared 86.94-99.9%". However, Fig 1b (y-axis; 1-ANI values) has many values below 86% (i.e., .18-.20).

We have used two datasets to obtain the ANI values.

1) Dataset II included *Microcoleus* strains sequenced in this study and the GenBank genomes. The ANI values shown in Fig. 1b correspond to the genomes from dataset II (refer to the Fig. 1 legend mentioning "dataset II"). To underscore the difference, we explicitly mentioned that the distance matrix used for this analysis has been based on dataset II (Line 574).

2) Dataset III included *Microcoleus* genomes sequenced in this study, and the numbers in Line 161 referred to only these genomes. We have now added clarification in this place (Line 162). They indeed have values between 86.94 and 99.9% (see Supplementary Data 5).

We thank the reviewer for this comment because a closer inspection of the ANI values used in Fig. 1b revealed that we had a few points that were erroneously placed closer to the 0.20 threshold. However, the lowest ANI value between genomes of dataset III was 82.4%. We have updated Figure 1b accordingly, and the previous observations, including the correlation analysis (Supplementary Data 13), have not been altered.

Line 210: "In particular, 1,079 genes showed significant associations, while 1,258 were significantly disassociating, .." how was significance determined?

The significance of (dis)association is an integral part of the software that was used to determine them. The program Coinfinder uses Bonferroni corrected test statistics to evaluate expected and observed rates of (dis)association, which allows it to determine genes that are significantly

(dis)associating (for more details [10.1099/mgen.0.000338](https://doi.org/10.1099/mgen.0.000338)). We have added a clarification from Line 199.

Line 341: *"We found no apparent peaks of elevated differentiation (neither 'islands' nor 'continents'; sensu Shapiro & Polz18) across the genome but rather a broad elevation of its diversity (Supplementary Fig. 8) as previously"*

Could this not be the result of them being different species instead of being different points in a continuum of speciation?

In this study, we are referring to species as groups of individuals and populations that are genetically and ecologically isolated from other such groups. In contrast, the speciation continuum represents a continuum of barriers to gene flow, where species can be at various points of genetic and ecological differentiation. Having only closely related individuals in our dataset, we could test whether they are at different points of the speciation continuum by investigating their genetic relatedness, the extent of gene flow, as well as levels of ecological and genetic differentiation. From our collective observations, we believe that our current dataset reflects a speciation continuum, where some lineages appear to be more and some less diverged (e.g., five lineages had a 93.2% probability of becoming fully separated species, and four had only a 73.2% probability of becoming separated species).

Typically, a broad elevation of diversity is observed at later stages of genetic differentiation; however, *Microcoleus* lineages had genome-wide mean pairwise F_{ST} from 0.2 to 0.9. This indicates that there is still a low level of differentiation between some lineages (e.g., M5 and M6), and they cannot be considered species yet (also considering the extent of gene flow between the lineages). Therefore, our dataset reveals *Microcoleus* lineages as appearing along the continuum of speciation, ranging from a low to a high probability of becoming fully diverged species.

Line 348: *"Given the heterogeneity of soil systems and their selective pressures on organisms, having small genomic regions ('islands') with few specific genes may be evolutionary disadvantageous." I do not follow this logic.*

In this part, we wanted to highlight that soil is a very dynamic and heterogeneous environment, often undergoing drastic oscillations in environmental parameters due to climatic shifts. Small genomic regions ('islands') usually have one or a few loci associated with some specific adaptation. As we have not observed such regions in our dataset, we hypothesized that for *Microcoleus*, it may be disadvantageous to have selection acting on one or few loci. Rather, positive selection acts on multiple loci scattered across the whole genome, which might be more advantageous so *Microcoleus* could rapidly respond to environmental stresses.

We have modified the sentence to improve clarity (Line 344-345).

Line 511: *"SNPs were detected by freebayes v1.3.299 with..."*

Given that the core genome is only 5% of the pan-genome or ~600 genes were SNPs from the flexible genomes used for the phylogenies?

Correct. We have used dataset III (genomes sequenced in this study) to construct species trees in three ways: 1) using the single-copy orthologues, 2) gene trees, and 3) all SNPs across the genomes. We have previously specified in the results sections (see Line 145) that all SNPs have been used, including core and flexible genome. We are now including this information in methods as well in order to be more precise (Line 535).

Line 540: "The best-fitting model selected by ModelFinder104 was LG+I+G and branch supports were computed using ultrafast bootstrapping with 2000..."

I'm not asking them to redo all of this but for this dataset specifically (which is the cyanobacterial phylogeny) site homogeneous models such as LG+ would not be as robust as a CAT or C60 (PMSF) model (site heterogeneous) especially when its 2000 single-copy orthologs and cyanobacterial are quite well known for their complicated phylogenies. Datasets 2 and 3 would probably be fine with LG+ models since they are all the same genus.

We thank the reviewer for the suggestion. While we recognize that different substitution models are expected to impact the resolution of phylogenies, we chose to be consistent and use ModelFinder to detect all models used in our phylogenetic analyses. Moreover, models like CAT-GTR have been criticized due to high computational demand and complexity. Ultimately, it has been shown that using these models for phylogenetic reconstruction often provides less reliable results compared to conventional models (e.g., 10.1093/sysbio/syw084).

Nevertheless, in response to the suggestion, we have performed an additional phylogenetic analysis on dataset I with the recommended PMSF substitution model (using ultrafast bootstrapping with 1000 replications) in IQ-TREE as it is much faster than other C models. The resulting topology did not change from the one shown in the original Fig. S1 (see the attached tree at the end of our responses to reviewers). In the attached tree, nodes without displayed support (blank nodes) are fully supported (100), and the midpoint root node has a support value of 0. Therefore, we have decided to keep the original Fig. S1 with a model detected by ModelFinder.

Line 578: The 16S phylogeny was not used for dating. How resolved was it? This might be a good thing to include in the supplementary data.

One of the reasons for using the phylogenetic tree based on the core genome alignment instead of the 16S rRNA was the low support of the 16S phylogeny. We have included the XML file for dating on GitHub to avoid further increasing the supplementary information.

Line 641: Is the assumption of a strict clock justified? Fig. S1 doesn't not look like it.

I did not find a link to 16S Microcoleus dataset used for the analyses in the manuscript, nor on the linked github and figshare sites.

The text says that the analysis used GTR+G+I. Is GTR appropriate in this case? I assume the

tree was calculated in iqtree, which by default provides a compositional analysis, and flags sequences with a significantly different composition.

We thank reviewer #3 for bringing up the question on the model, as we have realized that we actually used model HKY+I+G, not the GTR+G+I. ModelFinder determined this model as in all our previous phylogenetic analyses. We have now modified the sentence accordingly (Line 555), and updated GitHub with the XML file for dating, as mentioned above.

In response to the reviewer's mention of Fig. S1, we clarify that we did not perform the dating analysis based on dataset I (represented in Fig. S1) but on dataset III (see Lines 120-122 and 555). That would mean that dating has been performed on the genus level, not across all cyanobacterial taxa.

While Fig. S1 shows that a strict clock might not be suitable for investigating macroevolutionary patterns, it is important to note that dataset III included only closely related individuals within the genus *Microcoleus*. A strict clock is favored when considering taxa that emerged relatively recently (e.g., with a Miocene root; 10.1186/1471-2148-11-271). Additionally, a strict clock model is advantageous when the evolutionary rates do not appear to change significantly over time or among closely related lineages/species (10.1186/1471-2148-11-271).

On the other hand, the use of a relaxed molecular clock in our dataset would be computationally more demanding, and it would elevate uncertainties compared to a strict clock (HPD value bars would be significantly longer). Furthermore, using a relaxed molecular clock would demand investigation of additional parameters such as generation times and population sizes (e.g., 10.1099/mgen.0.000094), which extend beyond the scope of our current study. Relaxed clocks would have been better if more robust calibration points were established to date cyanobacterial evolution.

Considering our response to reviewer #3 along with the the one to reviewer #1, we have decided to keep our original analyses with a strict molecular clock and 16S rRNA. See a more detailed discussion on dating in the answer to reviewer #1.

Reviewer #4 (Remarks to the Author):

The new version of the manuscript from Stanojković and colleagues "The global speciation continuum of the cyanobacterium Microcoleus" has improved too much considering that authors took into account most of the main concerns of the reviewers. It is acknowledged that some claims are now more conservative and that the highly controversial topic of "species" was solved with the usage of the "lineages" word. The writing was benefited of shortening some sections and figures/figure legends were improved. The discussion section would benefit of a division in subsections with clear headings. Taking all of this into account, including the previous comments of this reviewer, this manuscript is suggested to be accepted to be published in the Nature Communications journal. However, there are some minor comments which should be addressed

but with no necessity of another round of revision according to this reviewer.

We thank reviewer #4 for their positive assessment and helpful suggestions that improved the manuscript. While we agree that dividing the discussion into subsections might be beneficial, we should comply with the formatting instructions of the journal and keep the discussion without subheadings.

Minor comments:

line 93 and elsewhere: please, be consistent with the number of sequenced strains. In some places appears 201 and in others 202.

Line 93 referred to the number of *Microcoleus* genomes sequenced in this study (without the outgroup). We have now changed the number to 202 and modified the sentence accordingly.

For instance, Lines 192 and 208 refer to the number of *Microcoleus* genomes without the outgroup. In the Results section (Lines 122-125), we highlighted the analyses using 201 genomes (without the outgroup). Different numbers of the genomes do not represent inconsistencies but rather reflect the nature of the data used for different analyses, outlined in the Results section.

lines 22 - 225: These differences could be product of the genome size differences. This, as being correlated by genome size in the previous phrases. Authors, could make the direct link watching the differences of number of HGT normalized by each 1000 genes or genome size and seeing if the differences disappear between these lineages. The same for the analyses in the figure 3d (also if the differences are retained, they could be more linked to environmental factors rather than to genome size).

We thank the reviewer for the suggestion. We normalized the number of HGTs by genome size and observed that significant differences between the lineages were retained. Therefore, we have updated Figure 3 and replaced the number of HGTs with "number of HGTs per Mb" (see Figure 3c and 3d) as well as Supplementary Data 10 with the Dunn test results.

Concerning these results, we have added one more sentence in the results section (Line 219-221) and modified the discussion accordingly (Line 396).

line 243: Supplementary Data 12?

Yes, we have corrected it now.

line 281: Please, fix the figure numbers.

Done.

line 297: Please fix Supplementary data number.

The Supplementary Data number was correct, but we added a further explanation to clarify it (Line 292).

line 314: Are you referring to Figs. 5a and 5b?

We were referring to Figs. 4a and 4b showing the extent of gene flow between and within the lineages.

line 337: "different species" maybe sounds better.

We have now added "different".

line 357: "Role" should be in plural.

Done.

line 373: Here are you referin just to the different species mentioned by the UPCEL analysis, or to the 12 lineages?.

We were referring to the 12 lineages. We have changed the sentence accordingly (Line 369).

line 421: RcsC if authors are referring to the protein.

Done.

line 492: Maybe are you referring to "high quality reads".

Yes, we are. We have modified the sentence accordingly.

Fig 1. I don't know if it is a problem of the pdf, but the squares on the habitats legend are not aligned.

The squares on the habitats' legends were not aligned. We fixed the issue, together with modifications explained in our response to reviewer #3.

Reviewer #3 (Remarks to the Author):

The authors addressed all of my concerns.